# Understanding Generalization in Physics Informed Models through Affine Variety Dimensions

## Abstract

In recent years, physics-informed machine learning has gained significant attention for its ability to enhance statistical performance and sample efficiency by integrating physical structures into machine learning models. These structures, such as differential equations, conservation laws, and symmetries, serve as inductive biases that can improve the generalization capacity of the hybrid model. However, the mechanisms by which these physical structures enhance generalization capacity are not fully understood, limiting the ability to guarantee the performance of the models. In this study, we show that the generalization performance of linear regressors incorporating differential equation structures is determined by the dimension of the associated affine variety, rather than the number of parameters. This finding enables a unified analysis of various equations, including nonlinear ones. We introduce a method to approximate the dimension of the affine variety and provide experimental evidence to validate our theoretical insights.

## 1. Introduction

In recent years, physics-informed machine learning (PIML) has garnered significant attention (Rai & Sahu, 2020; Karniadakis et al., 2021; Cuomo et al., 2022; Hao et al., 2022). PIML is a hybrid approach that integrates physical knowledge into machine learning models for tasks involving physical phenomena. The hybrid models can leverage physical structures such as differential equations (Raissi et al., 2019), conservation laws (Jagtap et al., 2020), and symmetries (Akhound-Sadegh et al., 2024) as inductive biases. This approach can potentially enhance sample efficiency and generalization capabilities. These models have been empirically applied to a wide range of phenomena, with successful applications including thrombus material properties (Yin et al., 2021), fluid dynamics (Cai et al., 2021a; Jin et al., 2021), turbulence (Wang et al., 2020), and heat transfer problems (Cai et al., 2021b). Despite these empirical successes, the impact of physical structures on the generalization capacity of models is primarily understood for linear equations or equations with specific regularity (Arnone et al., 2022; Doumèche et al., 2024a). This limited understanding hampers the ability to ensure the performance and reliability of these hybrid methods.

In this study, we theoretically analyze the generalization capacity of physics-informed linear regressors that incorporate the structure of differential equations. We show that the generalization capacity of these models is determined by the dimension of the affine variety associated with the differential equations, rather than the number of parameters. This novel perspective allows for a unified analysis of various equations, including nonlinear ones. To support our theoretical findings, we introduce a method for approximately calculating the dimension of the affine variety and provide extensive experimental validation. Our results demonstrate that even in scenarios with a large number of parameters relative to the amount of data, the physical structure reduces the intrinsic dimension of the hypothesis space and prevents overfitting, corroborating our theoretical findings.

Our paper is structured as follows. In Section 3, we outline the problem setup and present our main theoretical results, including a minimax risk analysis that underscores the role of the dimension of the affine variety. In Section 4, we discuss the dimension of affine variety especially in the context of nonlinear operators and introduce methods for their approximate calculation. Section 5 provides experimental evidence supporting our theoretical claims, demonstrating the practical advantages of incorporating physical structures in machine learning models.

## 2. Related Work

Since the seminal work by Raissi et al. (2019) on Physics-Informed Neural Networks (PINNs), PIML has rapidly emerged as a significant field of study. This area has been

---
[1]Anonymous Institution, Anonymous City, Anonymous Region, Anonymous Country. Correspondence to: Anonymous Author <anon.email@domain.com>.

Preliminary work. Under review by the International Conference on Machine Learning (ICML). Do not distribute.

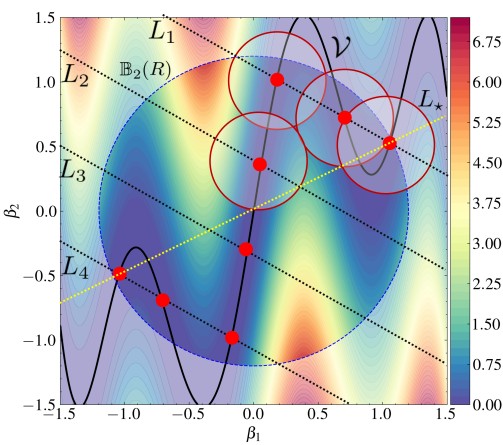

*Figure 1.* Illustration of the construction of the $\epsilon$-covering of the affine variety $\mathcal{V} \subseteq \mathbb{R}^2$ and the associated loss landscape. The black curve represents a $(K, d_\mathcal{V})$ regular affine variety with dimension $d_\mathcal{V} = 1$. The color gradients depict the loss landscape $\mathcal{L}(\boldsymbol{\beta}) := \sum_{k \in \mathbb{N}} \|p_k(\boldsymbol{\beta})\|_2^2$ of the equations defining $\mathcal{V} = \{\boldsymbol{\beta} : p_k(\boldsymbol{\beta}) = 0, k \in \mathbb{N}\}$. The blue dotted line represents a $\ell_2$ ball of radius $R$. The affine variety constrained with the $\ell_2$ ball is covered by $\epsilon$-balls centered at the intersections of $\mathcal{V}$ with four given subspaces $\{L_s\}_{s=1}^4$, shown as red points. The upper bound on the number of intersections of every subspace with the variety is $K$, while the actual maximum number is 5 formed by the subspace $L_\star$ (the yellow dotted line). The loss landscape of the equations is zero on $\mathcal{V}$ and locally convex around the points in $\mathcal{V}$.

comprehensively surveyed in the literature by (Rai & Sahu, 2020; Karniadakis et al., 2021; Cuomo et al., 2022; Hao et al., 2022). Leveraging the high function approximation capabilities of neural networks (Hornik et al., 1989; Kutyniok et al., 2022; De Ryck & Mishra, 2022), these models have been employed as versatile surrogates for solving various equations. In contrast, linear models are also used because of their interpretability, consistency with classical numerical solvers (Arnone et al., 2022; Ferraccioli et al., 2022), and the close relationship between Partial Differential Equations (PDEs) and kernel methods (Schaback & Wendland, 2006; Chen et al., 2021; Long et al., 2022; Dalton et al., 2024; Doumèche et al., 2024b). Recently, methods that exploit underlying conservation laws (Jagtap et al., 2020; Hu et al., 2022) and symmetries (Akhound-Sadegh et al., 2024; Dalton et al., 2024), in addition to the equations themselves, have also been developed.

Recent studies have made advances in the theoretical understanding of PINNs. Shin (2020) rigorously showed that the minimizer of the PINN loss converges to the strong solution as the data size approaches infinity for linear elliptic and parabolic PDEs under certain conditions. These findings were extended by Shin et al. (2023) into a general framework applicable to broader linear problems, with the loss function formulated in both strong and variational forms.

Mishra & Molinaro (2022; 2023) use the stability properties of the underlying PDEs to derive upper bounds on the generalization error of PINNs. Subsequent research has applied this analytical framework to various specific equations (Bai et al., 2021; Mishra & Molinaro, 2021). However, studies explicitly addressing the impact of physical structure on generalization capabilities are still limited. Arnone et al. (2022) proved that for second-order elliptic PDEs, the physics-informed linear estimator using a finite element basis converges at a rate surpassing the Sobolev minimax rate. Doumèche et al. (2024a) quantified the generalization capacity of the physics-informed estimator for general linear PDEs using the concept of effective dimension (Caponnetto & De Vito, 2007), a well-known metric in kernel method analysis. The effects of incorporating the structures of nonlinear complex equations, as well as conservation laws and symmetries, into models on generalization, have yet to be thoroughly analyzed.

## 3. Minimax risk Analysis

In this section, we explain how introducing physical structures can improve the generalization capacity of linear models. In Section 3.1, we provide preliminary knowledge about the affine variety. In Section 3.2, we outline the problem setup. In Section 3.3, we perform a minimax risk analysis, showing that the generalization capacity is mainly determined by the dimension of the affine variety. In Section 3.4, we show that our theory aligns with existing theories on linear operators.

### 3.1. Preliminaries on Affine Varieties

In this section, we provide the minimal background on affine varieties necessary for the subsequent sections. Let $\mathbb{K}[\boldsymbol{\beta}]$ denote the set of polynomials of the variables $\boldsymbol{\beta} = (\beta_1, \ldots, \beta_d) \in \mathbb{K}^d$ in the field $\mathbb{K}$. An affine variety $V(p_1, \ldots, p_K) \subseteq \mathbb{K}^d$ defined by the polynomials $p_1, \ldots, p_K \in \mathbb{K}[\boldsymbol{\beta}]$ is given by:

$$V(p_1, \ldots, p_K) := \left\{ \boldsymbol{\beta} \in \mathbb{K}^d : p_k(\boldsymbol{\beta}) = 0, \; \forall k \in [K] \right\},$$

where $[K] := \{1, \ldots, K\}$ is index set.

The dimension of an affine variety is defined as the maximal length $d_V$ of the chains $V_0 \subset V_1 \subset \ldots \subset V_{d_V}$ of distinct nonempty subvarieties of $V$. If the generating polynomials $\{p_k\}_{k=1}^K$ are all linear, the dimension of $V$ is defined as the maximal length of the increasing sequence of linear subspaces within $V$, which is the dimension of a variety $V$ as a linear space. For example, let $\mathbb{K} = \mathbb{R}$ and $V \subset \mathbb{R}^3$ is the plane: $V = \{(x, y, z) : x + y - z = 0\}$. A chain of subvarieties within $V$ is $V_0 \subset V_1 \subset V_2$, where $V_0 = \{(0, 0, 0)\}$ (a point, 0-dimensional), $V_1 = \{(t, 0, t) : t \in \mathbb{R}\}$ (a line, 1-dimensional), and $V_2 = V$ itself (the plane, 2-dimensional). The maximal length of the nested subvarieties

is two, i.e., $\dim(V) = 2$, which means that a plane has two degrees of freedom. Please refer to Appendix A for a precise definition of subvariety.

Next, we informally define the concept of a regular set for real affine varieties, which is used in Section 3.3 (for a formal definition, see Definition 2.1 in (Zhang & Kileel, 2023)).

A affine variety $V \subseteq \mathbb{R}^d$ is a $(K, d_V)$-regular set if:

1. For almost all affine planes $L$ with $\operatorname{codim}(L) \leq d_V$ in $\mathbb{R}^d$, $V \cap L$ has at most $K$ path-connected components.

2. For almost all affine planes $L$ with $\operatorname{codim}(L) > d_V$ in $\mathbb{R}^d$, $V \cap L$ is empty.

The notion $\operatorname{codim}$ represents the *codimension*. For an affine subspace $L \subseteq \mathbb{R}^d$, its codimension is defined by $\operatorname{codim}(L) = d - \dim(L)$. Simply put, codimension is how many dimensions you are "missing" when comparing a smaller space inside a bigger space.

A regular set restricts the complexity of a variety $V$. Intuitively, the complexity of $V$ can be measured by the number of connected components in its cross sections. For instance, a complex shape may have cross sections that split into multiple connected components. The larger the number of connected components $K$, the more complex the topology of $V$. Moreover, the dimension at which we slice the variety is also important. If the slice (affine plane) is large enough in dimension, i.e., the codimension is small ($< d_V$), then any intersection of the slice with $V$ is limited to at most $K$ connected pieces. Otherwise, the slice typically does not intersect $V$ at all. For example, consider the circle $V = \{(x, y) \in \mathbb{R}^2 \mid x^2 + y^2 - 1 = 0\}$. A line ($\operatorname{codim}(L) = 1$) intersects the circle in at most two points. For a single point ($\operatorname{codim}(L) = 2$), almost all points do not lie on the circle; that is, intersections with higher-codimension affine subspaces are almost empty. This implies the circle is a $(2, 1)$-regular set.

### 3.2. Problem Setup

**Formulation:** We address the regression problem, which aims to learn the unknown function $f^* \colon \mathbb{R}^m \to \mathbb{R}$ that satisfies the differential equation. We have a dataset consisting of $n$ observations, denoted as $\{(x_i, y_i)\}_{i=1}^n$, where $x_i \in \Omega$ represents the input within the input domain $\Omega \subseteq \mathbb{R}^m$ and $y_i \in \mathbb{R}$ represents the corresponding output. Observations are sampled independently from a probability distribution $\mathcal{P}$ on the domain $\Omega \times \mathbb{R}$. The relationship between the observations and the true function can be expressed as:

$$y_i = f^*(x_i) + \epsilon_i, \ \epsilon_i \sim \mathcal{N}(0, \sigma^2),$$

where $\epsilon_i$ represents normally distributed noise with mean zero and variance $\sigma^2$. The target function $f^*$ is the solution

of the differential equation, i.e., $\mathscr{D}[f^*] = 0$ for a given operator $\mathscr{D} \colon L^2(\Omega) \to L^2(\Omega)$, where $L^2(\Omega)$ denotes the space of square-integrable functions on a domain $\Omega \subseteq \mathbb{R}^m$.

To estimate the unknown function $f^*$, we consider an physics-informed regression problem for a hypothesis $\hat{f}_n \in \mathcal{H}$. Specifically, we require $\hat{f}_n$ to satisfy $\mathscr{D}[\hat{f}_n] = 0$ *in the weak sense*. The weak formulation bypasses the necessity for derivatives in the classical sense, instead requiring agreement in an integral sense using test functions. This approach relaxes the smoothness requirements for the solution. To define the weak formulation precisely, we set a set of pairs:

$$\mathcal{T} := \{(\psi_k, \mu_k)\}_{k \in \mathbb{N}},$$

where each $\psi_k \colon \mathbb{R}^m \to \mathbb{R}$ is a finite test function, and $\mu_k \colon \Sigma \to \mathbb{R}$ is a measure on the $\sigma$-algebra $\Sigma$ over the domain $\Omega$. A function $f$ is said to be a *weak solution* of the differential equation $\mathscr{D}[f] = 0$ if, for every pair $(\psi_k, \mu_k) \in \mathcal{T}$, it satisfies

$$\langle \mathscr{D}[f], \psi_k \rangle_{\mu_k} = \int_\Omega \mathscr{D}[f] \, \psi_k \, \mathrm{d}\mu_k = 0, \qquad (1)$$

where $\langle f, g \rangle_{\mu_k} = \int_\Omega fg \mathrm{d}\mu_k$ is the inner product with respect to the measure $\mu_k$ in the function space $L^2(\Omega, \mu_k)$.

By adopting a measure-based integral, we can handle a wider range of solutions in a unified manner. If the measure is a Borel measure, then it corresponds to a common weak solution. Alternatively, by setting the measure to a Dirac measure, it aligns with the framework used in PINNs.

The problem is formulated as follows:

$$\hat{f}_n = \operatorname*{arg\,min}_{f \in \mathcal{F}(\mathscr{D}, \mathcal{T})} \frac{1}{n} \sum_{i=1}^n |y_i - f(x_i)|^2 + \lambda_n \|f\|^2,$$

$$\mathcal{F}(\mathscr{D}, \mathcal{T}) := \{f : \langle \mathscr{D}[f], \psi_k \rangle_{\mu_k} = 0, \ \forall (\psi_k, \mu_k) \in \mathcal{T}\},$$
$$(2)$$

where $\lambda_n$ is a regularization parameter, and $\| \cdot \|$ is the standard $L^2$ norm with respect to the Lebesgue measure.

**Linear Hypothesis:** We focus our analysis on a linear hypothesis spanned by a basis $\mathcal{B} := \{\phi_j \colon \mathbb{R}^m \to \mathbb{R}\}_{j \in \mathbb{N}}$.

$$\mathcal{H} := \left\{ f : f(x) = \boldsymbol{\beta}^\top \boldsymbol{\phi}(x) = \sum_{j=1}^d \beta_j \phi_j(x), \ \phi_j \in \mathcal{B} \right\},$$

where $\boldsymbol{\beta} = [\beta_1, \beta_2, \ldots, \beta_d]^\top \in \mathbb{R}^d$ represents the coefficients to be estimated. The problem Eq. (2) is reduced to the physics-informed linear regression (PILR) given by

$$\hat{\boldsymbol{\beta}} = \operatorname*{arg\,min}_{\boldsymbol{\beta} \in \mathcal{V}(\mathscr{D}, \mathcal{B}, \mathcal{T})} \frac{1}{n} \|\boldsymbol{y} - \boldsymbol{\Phi}\boldsymbol{\beta}\|_2^2 + \lambda_n \|\boldsymbol{\beta}\|_2^2, \qquad (3)$$

$$\mathcal{V}(\mathscr{D}, \mathcal{B}, \mathcal{T}) :=$$
$$\left\{ \boldsymbol{\beta} : \langle \mathscr{D}[\boldsymbol{\beta}^\top \boldsymbol{\phi}], \psi_k \rangle_{\mu_k} = 0, \ \forall (\psi_k, \mu_k) \in \mathcal{T}, \phi_j \in \mathcal{B} \right\},$$
$$(4)$$

where $\boldsymbol{y} = [y_1, y_2, \ldots, y_n]^\top \in \mathbb{R}^n$ is the target vector, $\boldsymbol{\Phi} = [\boldsymbol{\phi}(x_1), \boldsymbol{\phi}(x_2), \ldots, \boldsymbol{\phi}(x_n)]^\top \in \mathbb{R}^{n \times d}$ is the design matrix, and $\|\cdot\|_2$ is the $\ell_2$-norm.

The set of coefficients $\mathcal{V}$ constitutes *an affine variety* as it represents the set of solutions to the $|\mathcal{T}|$ polynomial equations in the $d$ variables with real coefficients. For example, when $m = 1$ and $\mathscr{D}[f] = f \cdot \frac{\mathrm{d}}{\mathrm{d}x} f$, the affine variety $\mathcal{V}$ is defined by the solution set of the polynomial equations $p_k(\boldsymbol{\beta}) = \sum_{j,j'=1}^{d} \langle (\frac{\mathrm{d}}{\mathrm{d}x} \phi_j) \phi_{j'}, \psi_k \rangle_{\mu_k} \beta_j \beta_{j'} = 0$ for $k = 1, \ldots, |\mathcal{T}|$. We perform minimax risk analysis based on the dimension $d_\mathcal{V}$ of this affine variety because the affine variety $\mathcal{V}$ is crucial in determining the size of the intrinsic hypothesis space.

For simplicity, we define the equivalent formulation of Eq. (3) as follows.

$$\hat{\boldsymbol{\beta}} = \arg\min_{\boldsymbol{\beta} \in \mathcal{V}_R} \frac{1}{n} \|\boldsymbol{y} - \boldsymbol{\Phi}\boldsymbol{\beta}\|_2^2, \qquad (5)$$

where $\mathcal{V}_R = \mathcal{V}(\mathscr{D}, \mathcal{B}, \mathcal{T}) \cap \mathbb{B}_2(R)$ is the affine variety constrained with the $\ell_2$-ball $\mathbb{B}_2(R)$ with the radius $R > 0$.

**Minimax risk:** The goal of our analysis is to obtain the upper bound of the minimax risk for PILR in Eq. (3), which is defined by

$$\min_{\hat{\boldsymbol{\beta}}} \max_{\boldsymbol{\beta}^* \in \mathcal{V}_R} \|\hat{\boldsymbol{\beta}} - \boldsymbol{\beta}^*\|_2^2, \qquad (6)$$

where $\boldsymbol{\beta}^* \in \mathcal{V}_R$ is the optimal weight. We only concern the estimation error by assuming $f^* = \boldsymbol{\beta}^{*\top} \boldsymbol{\phi}$.

**We strongly recommend referring to the example in Section 5.1 to understand our problem setting intuitively.**

### 3.3. Main Theorem

We first introduce a unified bound on the covering number of an affine variety, as shown by Zhang & Kileel (2023), to measure the complexity of the affine variety $\mathcal{V}$.

**Lemma 3.1** (Zhang & Kileel (2023)). *Let $V \subset \mathbb{R}^d$ be a $(K, d_V)$-regular set in the ball $\mathbb{B}_2(R)$ with the radius $R$. Then for all $\epsilon \in (0, \mathrm{diam}(V)]$,*

$$\log \mathcal{N}(V, \epsilon, \|\cdot\|_2) \le d_V \log\left(\frac{2R d_V d}{\epsilon}\right) + \log 2K. \quad (7)$$

This upper bound is obtained by slicing the affine variety $V$ with subspaces $\{L_s\}_{s \in \mathbb{N}}$ within $\mathbb{R}^d$ and covering $V$ with balls centered at the intersections of $L_s$ and $V$, *i.e.*, $V \subset \bigcup_s \bigcup_{v \in V \cap L_s} \mathbb{B}_2(v; \epsilon)$. The covering for the two-dimensional case is illustrated in Fig. 1. The first term, $(2R d_V d/\epsilon)^{d_V}$, represents the number of subspaces $L_s$ needed to cover the entire space. It is mainly determined by the intrinsic dimension $d_V$ of the affine variety, although it

is still influenced by the ambient dimension $d$. The quantity $K$ in the second term denotes the number of the intersections between a single subspace $L$ and the variety $V$, and represents the covering number of $V \cap L$. Topologically, it corresponds to the Betti numbers of the affine variety, which informally represent the number of holes in $V$. The upper bound on the quantity $K$ is given, for example, by the Petrovskii-Oleinik-Milnor inequality (Petrovskii & Oleinik, 1949; Oleinik, 1951; Milnor, 1964). Specifically, an affine variety $V \cap \mathbb{B}_2(R)$ defined by polynomials $\{p_k\}_{k \in [K]}$ of maximum degree $\rho$ and the $\ell_2$-ball is $(\rho(2\rho - 1)^{d+1}, d_V)$-regular. This intuitively suggests that as the maximum degree of the polynomials increases, the topology of the affine variety grows more complex.

Next, we present the upper bound on the minimax risk. The complete statement and proof are provided in Appendix B.

**Theorem 3.2** (informal). *Let $\mathcal{V}(\mathscr{D}, \mathcal{B}, \mathcal{T})$ be the $(K, d_\mathcal{V})$-regular affine variety defined in Eq. (4). Suppose that the basis function is bounded by a constant, the minimum eigenvalue of the design matrix is restricted, and the stability condition for the estimator holds. For $\delta \in (0, 1)$, with probability $1 - \delta$, the minimax risk for PILR defined by Eq. (6) is bounded by*

$$\mathcal{O}\left( \sqrt{\frac{d_\mathcal{V} \log(d_\mathcal{V} d)}{n}} + \sqrt{\frac{\log 2K}{n}} + 2\sqrt{\frac{\log(2/\delta)}{n}} \right). \quad (8)$$

*Proof Sketch.* The proof involves two steps, the first of which is standard while the second step is specific to our problem. In the first step, we take advantage of the fact that the estimator minimizes the least squares loss on the set $\mathcal{V}_R$. Through several algebraic transformations, we upper bound the $\ell_2$ prediction error by a term that represents the supremum of a empirical random process in the metric space of the affine variety $(\mathcal{V}_R, \|\cdot\|_2)$, which have sub-Gaussian increments. In the second step, we calculate the supremum of the random process using the covering number of the affine variety $\mathcal{V}_R$, which is obtained from Lemma 3.1. We derive a tail bound on the basis of the Dudley's integral. $\square$

Theorem 3.2 suggests that the minimax risk is primarily determined by the intrinsic dimension $d_\mathcal{V}$ of the affine variety $\mathcal{V}$ rather than the number of the ambient dimension $d$ when $K$ is small. In the absence of physical structure, the covering number is $(4R/\epsilon)^d$, thus the minimax risk is $\mathcal{O}(\sqrt{d/n})$. When $d_\mathcal{V} \ll d$, the physical structure improves the convergence rate of the minimax risk. The method for calculating the intrinsic dimension $d_\mathcal{V}$ of the affine variety is discussed in Section 4.

To qualitatively estimate the impact of the second term, we discuss the case where the generalization capacity is determined by the local size of the hypothesis space induced

by the learning algorithm, such as gradient descent. When using gradient descent for the optimization of Eq. (3), the weights $\boldsymbol{\beta}$ are likely to be trapped in the path-connected component near the initial point because of the local convexity of the loss landscape, as illustrated in Fig. 1. In this situation, the intrinsic size of the hypothesis space can be estimated as follows. The first term of the covering number shown in Eq. (7) (and consequently in Eq. (8)) remains unchanged because it depends on the dimension. On the other hand, the second term becomes smaller because it focuses on fewer path-connected components. Therefore, we infer that the dimension of the affine variety $d_\mathcal{V}$ in the first term primarily contributes to the generalization capacity, especially when it is determined by the local size of the hypothesis space.

### 3.4. Analysis on Linear Operator

We discuss the special case where $\mathscr{D}$ is a linear operator. The second term in Eq. (8) vanishes because the Petrovskii-Oleinik-Milnor inequality indicates $K = 1$. Thus, the minimax risk is $\mathcal{O}\left(\sqrt{d_\mathcal{V} \log(d_\mathcal{V} d)/n}\right)$. Furthermore, the affine variety $\mathcal{V}$ is the solution set of a homogeneous system of linear equations. That is, the affine variety can be written as $\mathcal{V}(\mathscr{D}, \mathcal{B}, \mathcal{T}) = \{\boldsymbol{\beta} : \boldsymbol{D}\boldsymbol{\beta} = \boldsymbol{0}\}$ using the matrix $\boldsymbol{D} \in \mathbb{R}^{|\mathcal{T}| \times d}$ defined by $D_{k,j} \coloneqq \langle \mathscr{D}[\phi_j], \psi_k \rangle_{\mu_k}$. The affine variety is a linear subspace of dimension $d_\mathcal{V} = \dim \ker \boldsymbol{D}$. From the rank–nullity theorem, $d_\mathcal{V} = d - \operatorname{rank} \boldsymbol{D}$, indicating that the higher the rank of the matrix $\boldsymbol{D}$, the better the minimax risk of regression.

We show that our theory is consistent with existing theories. The effect of incorporating physical structure, represented by linear differential equations, on generalization has been analyzed within the framework of kernel methods by (Doumèche et al., 2024a;b). They argued that the physical structure smooths the kernel and reduces the effective dimension, leading to an improvement in the $\ell_2$ predictive error. We first present the definition of the physics-informed (PI) kernel.

**Definition 3.1** (PI kernel (Doumèche et al., 2024a;b)). *Let* $\mathcal{B} = \{\phi_j\}_{j \in \mathbb{N}}$ *be a basis and* $\mathcal{T} = \{(\psi_k, \mu)\}_{k \in \mathbb{N}}$ *be test functions and measure* $\mu$. *The PI kernel associated with the affine variety* $\mathcal{V}(\mathscr{D}, \mathcal{B}, \mathcal{T}) = \{\boldsymbol{\beta} : \boldsymbol{D}\boldsymbol{\beta} = \boldsymbol{0}\}$ *is defined by*

$$K_{\boldsymbol{M}}(x, y) = \langle \boldsymbol{M}^{-\frac{1}{2}} \phi(x), \boldsymbol{M}^{-\frac{1}{2}} \phi(y) \rangle_{\mathbb{R}^d},$$
$$\boldsymbol{M} \coloneqq \xi \boldsymbol{I} + \nu \boldsymbol{D}^\top \boldsymbol{G} \boldsymbol{D}, \tag{9}$$

*where* $\langle \cdot, \cdot \rangle_{\mathbb{R}^d}$ *is an inner product in the Euclidean space* $\mathbb{R}^d$, $\boldsymbol{I} \in \mathbb{R}^{|\mathcal{B}| \times |\mathcal{B}|}$ *is the identity matrix,* $\boldsymbol{G} \in \mathbb{R}^{|\mathcal{T}| \times |\mathcal{T}|}$ *is the matrix of the inner product of the test functions,* i.e., $G_{k,k'} = \langle \psi_k, \psi_{k'} \rangle_\mu$, $\boldsymbol{D} \in \mathbb{R}^{|\mathcal{T}| \times |\mathcal{B}|}$ *is the matrix defined by* $D_{k,j} = \langle \mathscr{D}[\phi_j], \psi_k \rangle_\mu$, *and* $\xi, \nu \geq 0$ *are weights hyperparameters for the* $L^2$ *regularization and loss of differential equation, respectively.*

When the PI kernel has parameters $\xi > 0$ and $\nu = 0$, the regularized regression problem with a reproducing kernel Hilbert space (RKHS) is the standard ridge regression. Note that Definition 3.1 extends the original definition to more general test functions. The original PI kernel uses basis functions as test functions, i.e., $\mathcal{T} = \mathcal{B} \times \{\mu\}$.

Doumèche et al. (2024b) showed the effective dimension $\mathcal{N}(\xi, \nu)$ of the PI kernel is evaluated above by a computable quantity as follows:

$$\mathcal{N}(\xi, \nu) \lesssim \sum_{\lambda \in \sigma(\boldsymbol{C}\boldsymbol{M}(\xi,\nu)^{-1}\boldsymbol{C})} \frac{1}{1 + \lambda^{-1}}, \tag{10}$$

where $\sigma(\cdot)$ is the set of eigenvalues of the matrix, $\boldsymbol{C} \in \mathbb{R}^{|\mathcal{B}| \times |\mathcal{B}|}$ is the matrix of the inner product of the basis function, i.e., $C_{j,j'} = \langle \phi_j, \phi_{j'} \rangle_\mu$. Next, we demonstrate the upper bound of the effective dimension of the PI kernel defined with the affine variety.

**Theorem 3.3.** *The effective dimension of the PI kernel associated with the affine variety* $\mathcal{V}(\mathscr{D}, \mathcal{B}, \mathcal{T}) = \{\boldsymbol{\beta} : \boldsymbol{D}\boldsymbol{\beta} = \boldsymbol{0}\}$ *with dimension* $d_\mathcal{V}$ *is upper bounded by*

$$\mathcal{N}(\xi, \nu) \lesssim \sum_{j=1}^{d_\mathcal{V}} \frac{1}{1 + \xi} + \sum_{j=d_\mathcal{V}}^{d} \frac{1}{1 + \xi + \nu \lambda_j} \leq \frac{d}{1 + \xi}.$$

*where* $\{\lambda_j\}_{j=d_\mathcal{V}}^{d}$ *are the eigenvalues of the matrix* $\boldsymbol{D}^\top \boldsymbol{G} \boldsymbol{D}$.

Theorem 3.3 indicates that as the dimension of the affine variety $d_\mathcal{V} = d - \operatorname{rank} \boldsymbol{D}$ decreases, the upper bound of the effective dimension of the PI kernel becomes smaller. The bound for the PI kernel is tighter compared to the bound for ridge regression $d/(1 + \xi)$, indicating that the physical structure improves the bound.

Therefore, our theory is consistent with the existing theory of the PI kernel. The PI kernel theory measures the complexity of the hypothesis space through the spectrum of the matrix $\boldsymbol{D}$ and the base kernel $\langle \phi(x), \phi(y) \rangle_{\mathbb{R}^d}$, restricting the target operator $\mathscr{D}$ to be linear. In contrast, our theory allows for the analysis of even nonlinear operators by considering only the dimension (the number of zero eigenvalues of $\boldsymbol{D}$) rather than the full spectrum.

## 4. On the Dimension of an Affine Variety

In general, the dimension of the affine variety $V = \{\boldsymbol{\beta} : p_k(\boldsymbol{\beta}) = 0, \forall k \in [K]\}$ has many equivalent definitions in addition to the one given in Section 3.1. In particular, the following statements are all equivalent.

**Definition 4.1.** *The maximal length* $d$ *of the chains* $V_0 \subset V_1 \subset \ldots \subset V_d$ *of distinct nonempty subvarieties of* $V$.

**Definition 4.2.** *The degree of the denominator of the Hilbert series of the affine variety* $V$.

**Definition 4.3.** *The maximal dimension of the tangent vector spaces at the non-singular points $U \subseteq V$ of the variety.*

$$d_V = \max_{\boldsymbol{\beta} \in U} \left( d - \text{rank} \begin{bmatrix} \nabla^\top p_1(\boldsymbol{\beta}) \\ \vdots \\ \nabla^\top p_K(\boldsymbol{\beta}) \end{bmatrix} \right),$$

Although Definition 4.1 clearly indicates that the dimension represents the complexity of the set $V$, it is difficult to calculate the dimension according to this definition. Definition 4.2 shows that the dimension represents the algebraic complexity of the polynomial ring. Definition 4.3 characterizes the dimension based on the local structure of the affine variety, making it suitable for numerical calculation as discussed in Section 4.2. It generalizes the rank-nullity theorem $d_V = d - \text{rank } \boldsymbol{D}$ in the linear case, as mentioned in Section 3.4. Details of the concepts associated with these definitions are given in Appendix A.

### 4.1. Lower Bound

We demonstrate that the dimension $d_{\mathcal{V}}$ of the affine variety can be characterized by the linear part of the operator $\mathscr{D}$.

**Theorem 4.1.** *Suppose the operator $\mathscr{D}$ can be decomposed as $\mathscr{D} = \mathscr{L} + \mathscr{F}$, where $\mathscr{L}$ is a non-zero linear differential operator and $\mathscr{F}$ is a nonlinear operator. Then, we have*

$$d_{\mathcal{V}(\mathscr{L})} \leq d_{\mathcal{V}(\mathscr{D})}.$$

*Proof.* The point $\boldsymbol{\beta} = \boldsymbol{0}$ lies on $\mathcal{V}(\mathscr{D})$, and if $\mathscr{L} \neq 0$, it is non-singular. The rank of the Jacobian of polynomials $p_k(\boldsymbol{\beta}) = \langle \mathscr{D}[\boldsymbol{\beta}^\top \boldsymbol{\phi}], \psi_k \rangle_{\mu_k}$ at $\boldsymbol{\beta} = \boldsymbol{0}$ is equal to $d - d_{\mathcal{V}(\mathscr{L})}$. By Definition 3, we have $d_{\mathcal{V}(\mathscr{L})} \leq d_{\mathcal{V}(\mathscr{D})}$. $\square$

Combining the result of Theorem 4.1 with Theorem 3.2 suggests that the nonlinear part $\mathscr{F}$ of the operator increases the affine variety dimension, having a negative effect on generalization. Furthermore, the dimension of the affine variety associated with the linear part $\mathscr{L}$ can be easily computed through the matrix rank. Therefore, the lower bound of the dimension of the affine variety associated with the nonlinear operator $\mathscr{D}$ can be readily determined, allowing us to estimate the minimum required amount of data $n$.

### 4.2. Numerical Calculation Method

According to Definition 4.2, the dimension of an affine variety is typically obtained by calculating the degree of the denominator of the Hilbert series, by using Gröbner bases. However, the worst-case time complexity of Buchberger's algorithm (Buchberger, 1976), which is the basic algorithm for computing Gröbner bases, is double exponential with respect to the number of variables $d$. This implies that its application to the target affine variety $\mathcal{V}(\mathscr{D}, \mathcal{B}, \mathcal{T})$ is

impractical. Therefore, on the basis of 4.3, we approximate $d_{\mathcal{V}}$ by sampling $\boldsymbol{\beta}_1^*, \ldots, \boldsymbol{\beta}_N^*$ from the affine variety $\mathcal{V}$ with some distribution and computing $\max_{\boldsymbol{\beta}^* \in \{\boldsymbol{\beta}_1^*, \ldots, \boldsymbol{\beta}_N^*\}} d - \text{rank} \left( \nabla^\top [p_1(\boldsymbol{\beta}^*), \ldots, p_K(\boldsymbol{\beta}^*)]^\top \right)$. When the operator $\mathscr{D}$ is nonlinear, we perform simulations using various boundary values and project the obtained solutions onto the basis $\mathcal{B}$ to sample $\boldsymbol{\beta}^* \in \mathcal{V}$. For the linear operator, the dimension does not depend on the weight $\boldsymbol{\beta}$, and the rank of the matrix $\boldsymbol{D}$ discussed in Section 3.4 precisely gives the dimension $d_{\mathcal{V}}$.

## 5. Experiments

We compared the performance of ridge regression (RR) and physics-informed linear regression (PILR) defined in Eq. (3) for several specific equations by varying the data size $n$ and the number of parameters $d$. For each equation, we train solutions for 10 different initial or boundary conditions determined randomly and plotted the mean and standard deviation of the mean squared error (MSE) in the test data. Details of the experimental setup are given in Appendix D.

When the operator $\mathscr{D}$ is linear, the estimator of PILR is given by Doumèche et al. (2024b), which solves the problem Eq. (3) approximately, as follows:

$$\hat{\boldsymbol{\beta}} = (\boldsymbol{\Phi}^\top \boldsymbol{\Phi} + n\boldsymbol{M})^{-1} \boldsymbol{\Phi}^\top \boldsymbol{y},$$

where $\boldsymbol{M}$ is the matrix defined in Eq. (9). For nonlinear equations, we use the Adam optimizer to minimize the loss function, which incorporates the differential equation constraints as a soft penalty. The hyperparameters $\xi, \nu$, which are the weight of the $L^2$-regularization and the differential equation constraint, are tuned by monitoring the MSE loss on the validation data. In Section 5.1, we focus on learning strong solutions, while in Section 5.2, we address learning numerical solutions. The experimental code is included in the supplementary material.

### 5.1. Learning Strong Solutions

In this section, we investigate the strong solutions of the classical harmonic oscillator and the diffusion equation with periodic boundary conditions. The solutions to these equations can be obtained analytically. Through these straightforward examples, we demonstrate both analytically and numerically that the generalization performance is determined by the dimension of the affine variety.

**Harmonic Oscillator:** The initial value problem of a harmonic oscillator $\mathscr{D}[y] = 0$ with spring constant $k$ and mass $m$ on the domain $\Omega = [0, T]$ is given by:

$$\mathscr{D}[y] = \frac{d^2}{dt^2} y + \frac{k}{m} y, \ y(0) = y_0, \ \frac{d}{dt} y(0) = v_0,$$

where $y_0$ and $v_0$ are the initial position and velocity, respectively. The solution to the initial value problem is analytically given by $y(t) = y_0 \cos(\omega t) + \frac{v_0}{\omega} \sin(\omega t)$, $\omega = \sqrt{k/m}$.

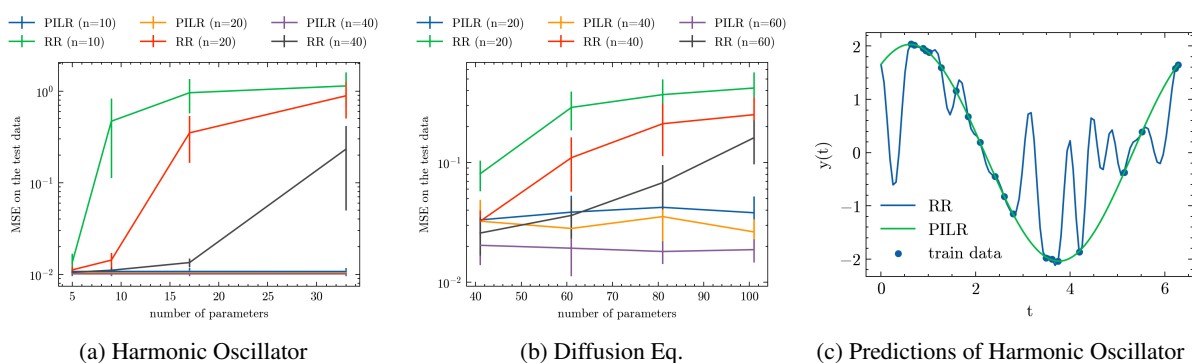

(a) Harmonic Oscillator        (b) Diffusion Eq.        (c) Predictions of Harmonic Oscillator

*Figure 2.* Experimental results for the strong solutions. (a, b) Test MSE (log scale) vs. number of parameters for the harmonic oscillator (a) and diffusion equation (b). The plots compare RR and PILR for three different data sizes $n$, showing the mean and standard deviation across 10 initializations. (c) Predictions of harmonic oscillator using a 33-parameter model trained on 20 samples: RR and PILR, with training data points indicated.

The settings for the basis functions and the test functions with the measure $\phi_j \in \mathcal{B}$, $(\psi_k, \mu_k) \in \mathcal{T}$ of indices $j = 1, \ldots, d_t$ and $k = 1, \ldots, K$ are as follows:

$$\phi_1(x) = 1, \ \phi_{2j}(x) = \cos(\omega_j x), \ \phi_{2j+1}(x) = \sin(\omega_j x),$$
$$\psi_k(x) = 1, \ \mu_k = \delta_{x_k},$$

where $\omega_j := \frac{j\pi}{T}$ is the $j$-th frequency and $\delta_{x_k}$ is the Dirac measure centered at the point $x_k \in \Omega$, which is uniformly sampled from data.

Then, the dimension of the affine variety is $d_\mathcal{V} = 2$, representing the essential degrees of freedom of the solution. Figure 2a supports our theory experimentally. For RR, the generalization performance degrades as the number of parameters $d = 2d_t + 1$ increases due to overfitting, as shown in Fig. 2c. In contrast, for PILR, the performance remains stable regardless of the number of parameters $d$ by virtue of the lower dimension of the affine variety $d_\mathcal{V} = 2$.

**Diffusion Equation:** The initial value problem for the one-dimensional diffusion equation $\mathscr{D}[u] = 0$ with diffusion coefficient $\alpha$ and periodic boundary conditions is given by:

$$\mathscr{D}[u] = \frac{\partial}{\partial t}u - \alpha\frac{\partial^2}{\partial x^2}u \qquad (x, t) \in [-\Xi, \Xi] \times [0, T]$$
$$u(x, 0) = u_0(x) \qquad\qquad x \in [-\Xi, \Xi]$$
$$u(-\Xi, t) = u(\Xi, t), \quad \frac{\partial u}{\partial x}(-\Xi, t) = \frac{\partial u}{\partial x}(\Xi, t).$$

The solution to the problem is analytically given by:

$$u(x, t) = \sum_{j=0}^{j_{\max}} [A_j\cos(\omega_j x) + B_j\sin(\omega_j x)]e^{-\alpha\omega_j^2 t},$$
$$A_j = \langle u_0, \cos(\omega_j x)\rangle, \ B_j = \langle u_0, \sin(\omega_j x)\rangle,$$

where $\omega_j := \frac{j\pi}{\Xi}$ is the $j$-th frequency. The maximum frequency of the initial value $u_0$ is set as $\omega_{\max} = j_{\max}\pi/\Xi$.

We define a basis functions combining the spatial Fourier basis and the time exponential function for indices $j = 0, \ldots, d_x$ and $j' = 0, \ldots, d_t$ as follows:

$$\phi_{2j,j'} = \cos(\omega_j x)e^{-\alpha\omega_{j'}^2 t}, \ \phi_{2j+1,j'} = \sin(\omega_j x)e^{-\alpha\omega_{j'}^2 t}.$$

The number of basis (number of parameters) is $d = 2d_x d_t + 1$, while the dimension of an affine variety is given by $d_\mathcal{V} = 2\min(d_x, d_t) + 1$. Figure 2b shows the results when we set $\alpha = 1.0$, $j_{\max} = 1$, $d_t = 2$, and vary $d_x$. The results indicate that the generalization performance of PILR does not deteriorate as $d_x$ increases, in contrast to RR.

### 5.2. Learning Numerical Solutions

In this section, we learn approximate solutions using numerical methods that use finite difference for four equations. In this setting, we consider the affine variety of the difference equation $\mathscr{D}_{\boldsymbol{h}}$ and the base functions $\mathcal{B}_{\boldsymbol{h}}$ and the test functions with the measure $\mathcal{T}_{\boldsymbol{h}}$ corresponding to the numerical method with step size $\boldsymbol{h}$. We first validate our theory using linear and nonlinear Bernoulli equations discretized by the explicit Euler method.

**Discrete Bernoulli Equation:** The discrete Bernoulli equation $\mathscr{D}_h[y] = 0$ with the step size $h$ on the domain $\Omega = [0, T]$ is given by

$$\mathscr{D}_h[y] = \frac{y_{\tau+1} - y_\tau}{h} + Py_\tau - Qy_\tau^\rho,$$

where $y_\tau = y(t_\tau)$ and $y_{\tau+1} = y(t_\tau + h)$ are evaluations on the $n_t$ size grid $\{t_\tau\}_{\tau=1}^{n_t}$, where $n_t := \frac{T}{h}$. The constant parameters $(P, Q, \rho)$ are set to $(1.0, 0.0, 0.0)$ for the linear case and to $(1.0, 0.5, 2)$ for the non-linear case. We use varying $n_t \in \{100, 200\}$ with $T = 1.0$ for both cases. The basis functions used correspond to the following one-dimensional piecewise constant functions of size $n_t$ for the Euler method, *i.e.*, $\phi_\tau(t) = 1$ for $t \in [t_\tau, t_{\tau+1})$ and 0 otherwise. The test

*Table 1.* Experimental results for the discrete linear and nonlinear Bernoulli equations approximated by the explicit Euler method. The settings include various step sizes $h$. The number of parameters (basis) $d$, and the calculated dimension of the affine variety $d_{\mathcal{V}}$.

| Settings | $\mathscr{D}_h$ | Linear Bernoulli eq. | | Nonlinear Bernoulli eq. | |
|---|---|---|---|---|---|
| | $h$ | $\frac{1}{100}$ | $\frac{1}{200}$ | $\frac{1}{100}$ | $\frac{1}{200}$ |
| Dimensions | $d$ | 100 | 200 | 100 | 200 |
| | $d_{\mathcal{V}}$ | 1 | 1 | 1 | 1 |
| Test MSE | RR | $0.48 \pm 0.32$ | $0.63 \pm 0.43$ | $0.60 \pm 0.41$ | $0.72 \pm 0.49$ |
| | PILR | $0.012 \pm 0.0025$ | $0.011 \pm 0.0013$ | $0.013 \pm 0.0024$ | $0.013 \pm 0.0018$ |

*Table 2.* Experimental results for the discrete linear and nonlinear diffusion equations approximated by the FDM. The settings include various step sizes $\boldsymbol{h} = (h_t, h_x)$. The number of parameters (basis) $d$, and the calculated dimension of the affine variety $d_{\mathcal{V}}$.

| Settings | $\mathscr{D}_h$ | Linear diffusion eq. | | | Nonlinear diffusion eq. | | |
|---|---|---|---|---|---|---|---|
| | $(h_t, h_x)$ | $\left(\frac{1}{400}, \frac{2}{10}\right)$ | $\left(\frac{1}{400}, \frac{2}{20}\right)$ | $\left(\frac{1}{400}, \frac{2}{30}\right)$ | $\left(\frac{1}{200}, \frac{2}{10}\right)$ | $\left(\frac{1}{200}, \frac{2}{20}\right)$ | $\left(\frac{1}{200}, \frac{2}{30}\right)$ |
| Dimensions | $d$ | 4010 | 8020 | 12030 | 2010 | 4020 | 6030 |
| | $d_{\mathcal{V}}$ | 10 | 20 | 30 | 10 | 20 | 30 |
| Test MSE | RR | $2.21 \pm 0.56$ | $2.14 \pm 0.57$ | $2.15 \pm 0.57$ | $1.12 \pm 0.40$ | $1.11 \pm 0.40$ | $1.12 \pm 0.40$ |
| | PILR | $1.13 \pm 0.30$ | $0.79 \pm 0.16$ | $0.57 \pm 0.11$ | $0.26 \pm 0.11$ | $0.22 \pm 0.10$ | $0.31 \pm 0.14$ |

functions are constant functions that output 1, and the measure used is the Dirac measure $\delta_{t_\tau}$ centered at the collocation points. The results are shown in Table 1. The computed $d_V$ is very small compared to $d$ and is independent of the choice of $h$. In all settings, PILR outperforms RR.

Next, we validate our theory using linear and nonlinear diffusion equations approximated by the finite difference method (FDM).

**Discrete Diffusion Equation:** The one-dimensional discrete diffusion equation $\mathscr{D}_{\boldsymbol{h}}[u] = 0$ with the step size $\boldsymbol{h} = [h_t, h_x]^\top$ and the diffusion coefficient $\alpha(u)$ on the domain $\Omega = [-\Xi, \Xi] \times [0, T]$ is given by:

$$\mathscr{D}_{\boldsymbol{h}}[u] = \frac{u_j^{\tau+1} - u_j^\tau}{h_t} - \alpha(u_j^\tau)\frac{u_{j+1}^\tau - 2u_j^\tau + u_{j-1}^\tau}{h_x},$$

where $u_j^\tau := u(x_j, t_\tau)$, $u_j^{\tau+1} := u(x_j, t_\tau + h_t)$, and $u_{j\pm1}^\tau := u(x_j \pm h_x, t_\tau)$ are evaluations on the $n_x \times n_t$ size grid $\{x_j\}_{j=1}^{n_x} \times \{t_\tau\}_{\tau=1}^{n_t}$, where $n_x := \frac{2\Xi}{h_x}$ and $n_t := \frac{T}{h_t}$. The periodic boundary condition is adopted in the spatial domain, *i.e.*, $u_{n_x+j}^\tau = u_j^\tau$ for any $j \in \mathbb{N}$. The diffusion coefficient $\alpha(u) = 1.0$ is used for the linear case and $\alpha(u) = 0.1/(1 + u^2)$ for the nonlinear case.

The basis functions used correspond to the following two-dimensional piecewise constant functions of size $n_t \times n_x$ for the FDM, *i.e.*, $\phi_{j,\tau}(x, t) = 1$ for $(x, t) \in [x_j, x_{j+1}] \times [t_\tau, t_{\tau+1}]$ and 0 otherwise. The test functions are constant functions that output 1, and the measure used is the Dirac measure $\delta_{(x_j, t_\tau)}$ centered at the collocation points. Table 2 shows that PILR achieves higher performance than RR for large values of $d$. While the dimension $d_{\mathcal{V}}$ is independent

of the time discretization step size in the Euler method, it depends on the spatial discretization step size in the FDM. Additionally, for the linear heat equation, we observe that while the underlying equation is the same as in the experiments of Section 5.1, the dimension of the affine variety changes due to the different discretization of the target equation and the use of different basis functions.

## 6. Conclusion

In this study, we presented a novel method for analyzing physics-informed models using affine varieties defined by differential equations. We showed that the generalization capacity of linear models incorporating physical structures is determined by the dimension of the associated affine variety, rather than by the number of parameters. Our findings align with existing theories on linear equations, providing a unified theoretical framework. In addition, we introduced a method for calculating the dimension of the affine variety and numerically confirm that this dimension is smaller than the number of parameters. Our experiments validate our theoretical findings, showing that the smaller dimension helps prevent overfitting even when the number of parameters is large. Our analysis is limited to linear models and does not address the optimization process when using gradient descent. Adapting our analysis to conservation laws (Jagtap et al., 2020; Hu et al., 2022) or Lie symmetries (Akhound-Sadegh et al., 2024; Dalton et al., 2024) is a promising direction. Extending our analysis to deep networks, such as PINNs, remains a challenge for future work.

## Impact Statements

This paper presents work whose goal is to advance the field of Machine Learning. There are many potential societal consequences of our work, none which we feel must be specifically highlighted here.

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

## A. Mathematical Background on Affine Varieties

In this section, we provide a formal definition of several concepts related to affine varieties and review the definition of the dimension of an affine variety, as briefly described in Section 4.

An affine variety is a fundamental concept in algebraic geometry. It is a subset of an affine space, defined as the solution set to a system of polynomial equations. Let $\mathbb{K}[\boldsymbol{\beta}]$ denote the set of polynomials in the variables $\boldsymbol{\beta} = (\beta_1, \ldots, \beta_d) \in \mathbb{K}^d$ over a field $\mathbb{K}$ (often $\mathbb{R}$ or $\mathbb{C}$). An affine variety $V(p_1, \ldots, p_K) \subseteq \mathbb{K}^d$ defined by the polynomials $p_1, \ldots, p_K \in \mathbb{K}[\boldsymbol{\beta}]$ is given by:

$$V(p_1, \ldots, p_K) := \left\{ \boldsymbol{\beta} \in \mathbb{K}^d : p_k(\boldsymbol{\beta}) = 0, \ \forall k \in [K] \right\}.$$

The geometry of an affine variety is determined by the set of all polynomials that "vanish" on $V$, i.e., those that become zero for every point in $V$. This set is called the ideal of the affine variety, denoted $I(V)$, and is defined as follows:

$$I(V) := \left\{ p \in \mathbb{K}[\boldsymbol{\beta}] : p(\boldsymbol{\beta}) = 0, \ \forall \boldsymbol{\beta} \in V \right\}.$$

The generating polynomial set $\{p_k\}_{k=1}^K$ of the affine variety $V$ is a subset of the ideal $I(V)$.

The coordinate ring over $V$, denoted $\mathbb{K}[V]$, is introduced to identify polynomials that yield the same values on the variety $V$. Specifically, $\mathbb{K}[V]$ is defined as the quotient of the polynomial ring $\mathbb{K}[\boldsymbol{\beta}]$ by the ideal $I(V)$, i.e., $\mathbb{K}[\boldsymbol{\beta}]/I(V)$. In the coordinate ring $\mathbb{K}[V] = \mathbb{K}[\boldsymbol{\beta}]/I(V)$, the difference between $p$ and $q$ vanishes on $V$, i.e., $p(\boldsymbol{\beta}) = q(\boldsymbol{\beta})$ for all $\boldsymbol{\beta} \in V$, or equivalently $p - q \in I(V)$. Thus, $p$ and $q$ are considered the same element. From another viewpoint, the coordinate ring $\mathbb{K}[V]$ can be considered as a set of polynomials not included in the ideal $I(V)$.

Based on the above definitions, we review the definition of the dimension $d_V$ of the affine variety.

### A.1. Geometric View

Considering the affine variety $V$ as an affine space, we can naturally define a subvariety as an "subset" of the variety that also satisfies polynomial equations. Let $q_1, \ldots, q_S$ be polynomials in a ring. Define $\langle q_1, \ldots, q_S \rangle$ as the smallest ideal generated by $q_1, \ldots, q_S$; that is, $\langle q_1, \ldots, q_S \rangle$ consists of all finite sums of the form $\sum_{i=1}^S r_i q_i$ where each $r_i$ is in the ring: $\langle q_1, \ldots, q_S \rangle = \{\sum_{i=1}^S r_i q_i\}$. A subvariety $U$ of $V$ is defined as the zero set of a subset ideal $\langle q_1, \ldots, q_S \rangle \subseteq \mathbb{K}[\boldsymbol{\beta}]/I(V)$ given by:

$$U := \left\{ \boldsymbol{\beta} \in \mathbb{K}^d : q_s(\boldsymbol{\beta}) = 0, \ \forall q_s \in \langle q_1, \ldots, q_S \rangle \right\}.$$

By using the concept of subvarieties, the dimension of an affine variety is defined as follows:

**Definition 4.1.** *The maximal length $d$ of the chains $V_0 \subset V_1 \subset \ldots \subset V_d$ of distinct nonempty subvarieties of $V$.*

This definition intuitively represents the size of $V$ by the maximal length of an increasing sequence of subspaces. If the generating polynomials $\{p_k\}_{k=1}^K$ are all linear, the dimension of $V$ is defined as the maximal length of an increasing sequence of linear subspaces within $V$, which corresponds to the dimension of $V$ as a linear space.

When we focus on the local structure, the following equivalent definition is obtained:

**Definition 4.3.** *The maximal dimension of the tangent vector spaces at the non-singular points $U \subseteq V$ of the variety.*

$$d_V = \max_{\boldsymbol{\beta} \in U} \left( d - \mathrm{rank} \begin{bmatrix} \nabla^\top p_1(\boldsymbol{\beta}) \\ \vdots \\ \nabla^\top p_K(\boldsymbol{\beta}) \end{bmatrix} \right),$$

From this definition, we can see that the dimension $d_V$ is a global quantity that summarizes the local linearized structure of the affine variety $V$ at a point.

### A.2. Algebraic View

The structure of an affine variety is determined by the ideal $I(V)$. Intuitively, the larger $I(V)$ is, the more polynomial constraints there are, which means that $V$ becomes smaller, and consequently, the coordinate ring $\mathbb{K}[V]$ also becomes

smaller. From this perspective, it is natural to expect a deep connection between the dimension of the coordinate ring $\mathbb{K}[V]$ (and similarly the ideal $I(V)$) and the dimension of the affine variety $V$.

To explore this connection, we first discuss the dimension of the coordinate ring $\mathbb{K}[V]$ using Krull dimension. The ideal $\mathfrak{p} \subset R$ in a polynomial ring $R$ is prime if $\forall a, b \in R$, $ab \in \mathfrak{p} \Rightarrow a \in \mathfrak{p}$ or $b \in \mathfrak{p}$. The definition of the dimension of the affine variety through the Krull dimension is shown below.

**Definition A.1.** *The Krull dimension of the coordinate ring $\mathbb{K}[V]$: The maximum length $d$ of the chain of prime ideals $\mathfrak{p}_0 \subset \mathfrak{p}_1 \subset \cdots \subset \mathfrak{p}_d$ in the coordinate ring $\mathbb{K}[V]$.*

This definition signifies that the dimension of an affine variety is characterized in the world of polynomial sets by the maximal length of an increasing chain of "subsets" within the coordinate ring, corresponding to Definition 4.1 from a geometric perspective.

In contrast, the size of the coordinate ring $\mathbb{K}[V]$ can also be measured using Hilbert series. First, by homogenizing the defining equations by adding one variable $\gamma \in \mathbb{K}$, we embed the affine variety $V \subset \mathbb{K}^d$ into the projective variety $P \subset \mathbb{K}^{d+1}$. The projective variety $P(h_1, \ldots, h_K) \subset \mathbb{K}^{d+1}$, defined by the homogeneous polynomials $h_1, \ldots, h_K \in \mathbb{K}[(\boldsymbol{\beta}, \gamma)]$, is given by:

$$P(h_1, \ldots, h_K) := \left\{ (\boldsymbol{\beta}, \gamma) \in \mathbb{K}^{d+1} : h_k(\boldsymbol{\beta}, \gamma) = 0, \ \forall k \in [K] \right\}.$$

The dimension of the variety is also increased by one, i.e., $d_P = d_V + 1$. The coordinate ring $\mathbb{K}[P] = \mathbb{K}[(\boldsymbol{\beta}, \boldsymbol{\gamma})]/I(P)$ of the projective variety $P$ can be decomposed into subgroups (called the graded coordinate ring) as follows:

$$\mathbb{K}[P] = \bigoplus_{\rho \in \mathbb{N}} S_\rho, \ S_0 = \mathbb{K},$$

where $S_\rho$ is the set of homogeneous polynomials of degree $\rho$ modulo the ideal $I(P)$. As a metric for the size of the coordinate ring $\mathbb{K}[P]$, the Hilbert function $\mathrm{H}(\rho)$ and Hilbert-Poincaré series $\mathrm{HS}(t)$ are defined as follows:

$$\mathrm{H}(\rho) = \dim S_\rho, \ \mathrm{HS}(t) = \sum_{\rho \in \mathbb{N}} \mathrm{H}(\rho) t^\rho = \frac{\prod_{k=1}^K (1 - t^{\rho_k})}{(1 - t)^{d+1}},$$

where $\dim$ denotes the Krull dimension and $\rho_1, \ldots, \rho_K$ are the degrees of the homogeneous polynomials $h_1, \ldots, h_K$.

The Hilbert function represents the dimension of a "subspace" of the decomposed coordinate ring, and the Hilbert series is the generating function of the sequence of the Hilbert function, which is also a rational function with a pole at $t = 1$. These measures indicate the growth of the dimension of the homogeneous components of the algebra with respect to the degree. According to the dimension theorem, the Krull dimension of the projective variety $P$ matches the order of the Hilbert series at the pole $t = 1$, which is one of the most important results in commutative algebra.

Therefore, the dimension of the affine variety is defined using the Hilbert series, as follows:

**Definition 4.2.** *The degree of the denominator of the Hilbert series of the affine variety $V$.*

Given the Gröbner basis of the ideal $I(P)$, the Hilbert series can be easily computed, leading to an efficient estimation of the dimension of the affine variety $d_V$.

## B. Proof for Theorem 3.2

We first provide the assumptions:

**Assumption 1** (Boundedness of basis functions). *For the basis function $\boldsymbol{\phi} = [\phi_1, \ldots, \phi_d]^\top$, where $\phi_j \in \mathcal{B}$, there exists a positive constant $M$ such that $\|\boldsymbol{\phi}(x)\|_2 \leq M$ for all $x \in \Omega$.*

**Assumption 2** (Restricted lower eigenvalues). *There exists a constant $\kappa > 0$ such that $\frac{1}{\sqrt{n}}\|\boldsymbol{\Phi}\boldsymbol{\beta}\|_2 \geq \sqrt{\kappa}\|\boldsymbol{\beta}\|_2$ for all $\boldsymbol{\beta} \in \mathbb{R}^d$.*

**Assumption 3** (Stability of estimator). *There exists a constant $\Gamma > 1$ such that $\|\hat{\boldsymbol{\beta}}_1 - \hat{\boldsymbol{\beta}}_2\|_2 \leq (\Gamma - 1)\|\boldsymbol{\beta}_1^* - \boldsymbol{\beta}_2^*\|_2$, for the estimators $\hat{\boldsymbol{\beta}}_1$ and $\hat{\boldsymbol{\beta}}_2$ of the optimal weights $\boldsymbol{\beta}_1^*$ and $\boldsymbol{\beta}_2^*$, respectively.*

The following is the formal statement of Theorem 3.2.

**Theorem B.1** (Formal Statement of Theorem 3.2). *Let $\mathcal{V}(\mathscr{D}, \mathcal{B}, \mathcal{T})$ be the $(K, d_\mathcal{V})$–regular affine variety. Suppose Assumptions 1-3 hold. Under these assumptions, for $\delta \in (0, 1)$, with probability $1 - \delta$, the minimax risk for PILR is bounded as follows:*

$$\min_{\hat{\boldsymbol{\beta}}} \max_{\boldsymbol{\beta}^* \in \mathcal{V}_R} \|\hat{\boldsymbol{\beta}} - \boldsymbol{\beta}^*\|_2^2 \leq C \kappa^{-1} \sigma M \Gamma R \left( \sqrt{\frac{d_\mathcal{V} \log(d_\mathcal{V} d)}{n}} + \sqrt{\frac{\log 2K}{n}} + 2\sqrt{\frac{\log(2/\delta)}{n}} \right), \tag{11}$$

*where $C$ is a constant.*

*Proof.* **Step 1:** We first upper bound the prediction error by a term that represents the supremum of a empirical process in the metric space of the affine variety. Using Lemma B.2, we get:

$$\|\boldsymbol{\Phi}(\boldsymbol{\beta}^* - \hat{\boldsymbol{\beta}})\|_2^2 \leq 2\boldsymbol{\epsilon}^\top \boldsymbol{\Phi}(\boldsymbol{\beta}^* - \hat{\boldsymbol{\beta}}).$$

We denote $\mathrm{x}_\beta := \boldsymbol{\epsilon}^\top \boldsymbol{\Phi}(\boldsymbol{\beta} - \hat{\boldsymbol{\beta}})$ as the random process in the metric space $(\mathcal{V}_R, \|\cdot\|_2)$. Note that the estimator $\hat{\boldsymbol{\beta}}$ is a random variable depending on the parameter $\boldsymbol{\beta}$ and the noise $\boldsymbol{\epsilon}$. Then, the minimax risk is bounded as follows.

$$\min_{\hat{\boldsymbol{\beta}}} \max_{\boldsymbol{\beta}^* \in \mathcal{V}_R} \|\hat{\boldsymbol{\beta}} - \boldsymbol{\beta}^*\|_2^2 \leq \min_{\hat{\boldsymbol{\beta}}} \max_{\boldsymbol{\beta}^* \in \mathcal{V}_R} \frac{\kappa^{-1}}{n} \|\boldsymbol{\Phi}(\hat{\boldsymbol{\beta}} - \boldsymbol{\beta}^*)\|_2^2 \leq \frac{2}{n} \kappa^{-1} \sup_{\boldsymbol{\beta} \in \mathcal{V}_R} \mathrm{x}_\beta. \tag{12}$$

The first inequality holds by Assumption 2.

**Step 2:** Next, we calculate the supremum of the empirical process $\mathrm{x}_\beta$ using the covering number. For all $\boldsymbol{\beta}_1, \boldsymbol{\beta}_2 \in \mathcal{V}_R$, it is shown that the variable $\mathrm{x}_{\boldsymbol{\beta}_1} - \mathrm{x}_{\boldsymbol{\beta}_2}$ has sub-Gaussian increments with respect to the metric $\|\cdot\|_2$:

$$\begin{aligned} \mathrm{x}_{\boldsymbol{\beta}_1} - \mathrm{x}_{\boldsymbol{\beta}_2} &= \sum_{i=1}^n \epsilon_i ((\boldsymbol{\beta}_1 - \hat{\boldsymbol{\beta}}_1) - (\boldsymbol{\beta}_2 - \hat{\boldsymbol{\beta}}_2))^\top \phi(x_i) \\ &\leq \sum_{i=1}^n \epsilon_i \|(\boldsymbol{\beta}_1 - \boldsymbol{\beta}_2) - (\hat{\boldsymbol{\beta}}_1 - \hat{\boldsymbol{\beta}}_2)\|_2 \|\phi(x_i)\|_2 \\ &\leq \sum_{i=1}^n \epsilon_i \left( \|\boldsymbol{\beta}_1 - \boldsymbol{\beta}_2\|_2 + \|\hat{\boldsymbol{\beta}}_1 - \hat{\boldsymbol{\beta}}_2\|_2 \right) M \\ &\leq \Gamma \|\boldsymbol{\beta}_1 - \boldsymbol{\beta}_2\|_2 M \mathrm{e}, \end{aligned} \tag{13}$$

where e is the zero-mean Gaussian random variable with variance $n\sigma^2$. The second inequality holds by the Cauchy-Schwarz inequality and the third holds by the triangle inequality and Assumption 1. The last inequality holds by Assumption 3.

From Eq. (13), the random process $\mathrm{x}_{\boldsymbol{\beta}_1} - \mathrm{x}_{\boldsymbol{\beta}_2}$ has sub-Gaussian increments as follows.

$$\|\mathrm{x}_{\boldsymbol{\beta}_1} - \mathrm{x}_{\boldsymbol{\beta}_2}\|_{\psi_2} \leq \sqrt{n} \sigma M \Gamma \|Z\|_{\psi_2} \|\boldsymbol{\beta}_1 - \boldsymbol{\beta}_2\|_2,$$

where $Z$ is the standard Gaussian random variable and $\|\cdot\|_{\psi_2}$ is the sub-Gaussian norm. For the centered random process $\mathrm{z}_\beta := \mathrm{x}_\beta - \mathbb{E}[\mathrm{x}_\beta]$, $\|\mathrm{z}_{\boldsymbol{\beta}_1} - \mathrm{z}_{\boldsymbol{\beta}_2}\|_{\psi_2} \lesssim \|\mathrm{x}_{\boldsymbol{\beta}_1} - \mathrm{x}_{\boldsymbol{\beta}_2}\|_{\psi_2}$ holds because $\|\mathrm{x}_{\boldsymbol{\beta}_1} - \mathrm{x}_{\boldsymbol{\beta}_2}\|_{\psi_2}$ is sub-Gaussian.

Using Lemma B.3, we obtain the following bound with some constant $C_0$:

$$\mathbb{E} \sup_{\boldsymbol{\beta} \in \mathcal{V}_R} \mathrm{z}_\beta \leq C_0 \sqrt{n} \sigma M \Gamma R \left( \sqrt{d_\mathcal{V} \log d_\mathcal{V} d} + \sqrt{\log 2K} \right). \tag{14}$$

Next, using Dudley's integral tail bound, we have:

$$\mathbb{P} \left( \sup_{\boldsymbol{\beta} \in \mathcal{V}_R} \mathrm{z}_\beta \leq \mathbb{E} \sup_{\boldsymbol{\beta} \in \mathcal{V}_R} \mathrm{z}_\beta + C_0 \sqrt{n} \sigma M \Gamma 2R \sqrt{\log(\delta/2)} \right) \geq 1 - \delta.$$

By incorporating the non-centered process $\mathrm{x}_{\boldsymbol{\beta}}$, we obtain:

$$\mathbb{P}\left(\sup_{\boldsymbol{\beta}\in\mathcal{V}_R}\mathrm{x}_{\boldsymbol{\beta}} \leq \sup_{\boldsymbol{\beta}\in\mathcal{V}_R}|\mathbb{E}[\mathrm{x}_{\boldsymbol{\beta}}]| + \mathbb{E}\sup_{\boldsymbol{\beta}\in\mathcal{V}_R}\mathrm{z}_{\boldsymbol{\beta}} + C_0\sqrt{n}\sigma M\Gamma 2R\sqrt{\log(\delta/2)}\right) \geq 1-\delta. \tag{15}$$

To bound $\mathbb{E}[\mathrm{x}_{\boldsymbol{\beta}}]$, we note that:

$$\begin{aligned}
\mathbb{E}[\mathrm{x}_{\boldsymbol{\beta}}] &= \mathbb{E}\left[\boldsymbol{\epsilon}^{\top}\boldsymbol{\Phi}\left(\boldsymbol{\beta}-\hat{\boldsymbol{\beta}}\right)\right]\\
&= \mathbb{E}\left[\boldsymbol{\epsilon}^{\top}\boldsymbol{\Phi}\hat{\boldsymbol{\beta}}\right]\\
&\leq \sqrt{\mathbb{E}\left[\|\boldsymbol{\epsilon}^{\top}\boldsymbol{\Phi}\|_2^2\right]}\sqrt{\mathbb{E}\left[\left\|\hat{\boldsymbol{\beta}}\right\|_2^2\right]}\\
&\leq \sigma\sqrt{\sum_{j=1}^{d}\sum_{i=1}^{n}|\phi_j(x_i)|^2}R\\
&= \sigma\sqrt{n}MR
\end{aligned} \tag{16}$$

Here, the third inequality follows from the Cauchy-Schwarz inequality, and the fourth inequality is derived from the fact that $|\boldsymbol{\epsilon}^{\top}\boldsymbol{\Phi}_j|^2/(\sigma\|\boldsymbol{\Phi}_j\|_2)^2$ follows a chi-squared distribution with 1 degrees of freedom and $\hat{\boldsymbol{\beta}}\in\mathcal{V}_R$.

By combining Eq. (14), Eq. (15), and Eq. (16), we obtain the following bound with some constant $C$:

$$\mathbb{P}\left(\frac{2}{n}\sup_{\boldsymbol{\beta}\in\mathcal{V}_R}\mathrm{x}_{\boldsymbol{\beta}} \leq C\sigma M\Gamma R\left(\sqrt{\frac{d_{\mathcal{V}}\log d_{\mathcal{V}}d}{n}} + \sqrt{\frac{\log 2K}{n}} + 2\sqrt{\frac{\log(\delta/2)}{n}}\right)\right) \geq 1-\delta.$$

This completes the proof. $\qquad\square$

**Lemma B.2.** *Let $\hat{\boldsymbol{\beta}}$ be a minimizer of the following optimization problem:*

$$\hat{\boldsymbol{\beta}} = \arg\min_{\boldsymbol{\beta}\in\mathcal{V}_R}\frac{1}{n}\|\boldsymbol{y}-\boldsymbol{\Phi}\boldsymbol{\beta}\|_2^2, \tag{17}$$

*where $\mathcal{V}_R = \mathcal{V}(\mathcal{D},\mathcal{B},\mathcal{T})\cap\mathbb{B}_2(R)$ is the affine variety constrained with the $\ell_2$-ball, $\boldsymbol{y} = \boldsymbol{\Phi}\boldsymbol{\beta}^* + \boldsymbol{\epsilon}$ is the observed vector, $\boldsymbol{\Phi}$ is the design matrix, $\boldsymbol{\beta}^*\in\mathcal{V}_R$ is the true parameter vector, and $\boldsymbol{\epsilon} = [\epsilon_1,\ldots,\epsilon_n]^{\top}$ is the noise vector with each $\epsilon_i@$ independently following a zero-mean Gaussian distribution. Then, under these conditions, we have:*

$$\|\boldsymbol{\Phi}(\boldsymbol{\beta}^*-\hat{\boldsymbol{\beta}})\|_2^2 \leq 2\boldsymbol{\epsilon}^{\top}\boldsymbol{\Phi}(\boldsymbol{\beta}^*-\hat{\boldsymbol{\beta}}). \tag{18}$$

*Proof.* Since $\hat{\boldsymbol{\beta}}$ is a minimizer of Eq. (17), we have:

$$\|\boldsymbol{y}-\boldsymbol{\Phi}\hat{\boldsymbol{\beta}}\|_2^2 \leq \|\boldsymbol{y}-\boldsymbol{\Phi}\boldsymbol{\beta}^*\|_2^2 = \|\boldsymbol{\epsilon}\|_2^2.$$

The left-hand side can be expanded as:

$$\begin{aligned}
\|\boldsymbol{y}-\boldsymbol{\Phi}\hat{\boldsymbol{\beta}}\|_2^2 &= \|\boldsymbol{y}-\boldsymbol{\Phi}\boldsymbol{\beta}^* + \boldsymbol{\Phi}\boldsymbol{\beta}^* - \boldsymbol{\Phi}\hat{\boldsymbol{\beta}}\|_2^2\\
&= \|\boldsymbol{\epsilon}-\boldsymbol{\Phi}(\boldsymbol{\beta}^*-\hat{\boldsymbol{\beta}})\|_2^2.
\end{aligned}$$

Thus, we have:

$$\|\boldsymbol{\epsilon}-\boldsymbol{\Phi}(\boldsymbol{\beta}^*-\hat{\boldsymbol{\beta}})\|_2^2 \leq \|\boldsymbol{\epsilon}\|_2^2.$$

Expanding the left-hand side, we get:

$$\|\boldsymbol{\epsilon}-\boldsymbol{\Phi}(\boldsymbol{\beta}^*-\hat{\boldsymbol{\beta}})\|_2^2 = \|\boldsymbol{\epsilon}\|_2^2 - 2\boldsymbol{\epsilon}^{\top}\boldsymbol{\Phi}(\boldsymbol{\beta}^*-\hat{\boldsymbol{\beta}}) + \|\boldsymbol{\Phi}(\boldsymbol{\beta}^*-\hat{\boldsymbol{\beta}})\|_2^2.$$

Subtracting $\|\boldsymbol{\epsilon}\|_2^2$ from both sides, we obtain:

$$\|\boldsymbol{\Phi}(\boldsymbol{\beta}^*-\hat{\boldsymbol{\beta}})\|_2^2 \leq 2\boldsymbol{\epsilon}^{\top}\boldsymbol{\Phi}(\boldsymbol{\beta}^*-\hat{\boldsymbol{\beta}}).$$

This completes the proof. $\qquad\square$

**Lemma B.3.** *Let $z_{\boldsymbol{\beta}}$ be the zero-mean random process in the metric space $(\mathcal{V}_R, \|\cdot\|_2)$, which have the following sub-Gaussian increments. For all $\boldsymbol{\beta}_1, \boldsymbol{\beta}_2 \in \mathcal{V}_R$,*

$$\|z_{\boldsymbol{\beta}_1} - z_{\boldsymbol{\beta}_2}\|_{\psi_2} \leq A\|\boldsymbol{\beta}_1 - \boldsymbol{\beta}_2\|_2,$$

*where $\|\cdot\|_{\psi_2}$ is the sub-Gaussian norm, $A$ is a positive constant. Then, the expectation of the supremum of the process can be bounded as follows.*

$$\mathbb{E} \sup_{\boldsymbol{\beta} \in \mathcal{V}_R} z_{\boldsymbol{\beta}} \leq CAR\left(\sqrt{d_{\mathcal{V}} \log d_{\mathcal{V}} d} + \sqrt{\log 2K}\right),$$

*where $C$ is positive constant.*

*Proof.* Using Dudley's integral inequality (Dudley, 1967) to the zero-mean random process:

$$\mathbb{E} \sup_{\boldsymbol{\beta} \in \mathcal{V}_R} z_{\boldsymbol{\beta}} \leq C_0 A \int_0^{\infty} \sqrt{\log \mathcal{N}(\mathcal{V}_R, \epsilon, \|\cdot\|_2)} \mathrm{d}\epsilon. \tag{19}$$

Since the set $\mathcal{V}_R$ is $(K, d_{\mathcal{V}})$ regular set from Lemma 2.13 by Zhang & Kileel (2023), Lemma 3.1 shows the upper bound of the covering number for any $\epsilon \in (0, 2R]$ as follows.

$$\log \mathcal{N}(\mathcal{V}_R, \epsilon, \|\cdot\|_2) \leq d_{\mathcal{V}} \log\left(\frac{2R d_{\mathcal{V}} d}{\epsilon}\right) + \log 2K.$$

We substitute the above inequality to Eq. (19):

$$\mathbb{E} \sup_{\boldsymbol{\beta} \in \mathcal{V}_R} z_{\boldsymbol{\beta}} \leq C_0 A \left(\sqrt{d_{\mathcal{V}}} \int_0^{\infty} \sqrt{\log\left(\frac{2R d_{\mathcal{V}} d}{\epsilon}\right)} \mathrm{d}\epsilon + 2R\sqrt{\log 2K}\right).$$

The integral in the first term can be calculated using substitution and integration by parts. Let

$$I := \int_0^{\infty} \sqrt{\log\left(\frac{2R d_{\mathcal{V}} d}{\epsilon}\right)} \mathrm{d}\epsilon = \int_0^{2R} \sqrt{\log\left(\frac{2R d_{\mathcal{V}} d}{\epsilon}\right)} \mathrm{d}\epsilon.$$

We substitute $\alpha := 2R d_{\mathcal{V}} d$, $u := \log(\alpha/\epsilon)$ into the integral:

$$I = \int_{\infty}^{\log d_{\mathcal{V}} d} u^{1/2}(-\alpha e^{-u}) \mathrm{d}u.$$

To solve the above integral, we use the formula for integration by parts:

$$I = -\alpha \left([-u^{1/2}e^{-u}]_{\infty}^{\log d_{\mathcal{V}} d} + \frac{1}{2} \int_{\infty}^{\log d_{\mathcal{V}} d} u^{-1/2}e^{-u} \mathrm{d}u\right)$$

$$= 2R\sqrt{\log d_{\mathcal{V}} d} + R d_{\mathcal{V}} d \int_{\log d_{\mathcal{V}} d}^{\infty} u^{-1/2}e^{-u} \mathrm{d}u.$$

The integral in the second term can be upper bounded as follows.

$$\int_{\log d_{\mathcal{V}} d}^{\infty} u^{-1/2}e^{-u} \mathrm{d}u \leq \int_{\log d_{\mathcal{V}} d}^{\infty} e^{-u} \mathrm{d}u = [-e^{-u}]_{\log d_{\mathcal{V}} d}^{\infty} = (d_{\mathcal{V}} d)^{-1}.$$

We obtain the following bound with some constant $C$.

$$\mathbb{E} \sup_{\boldsymbol{\beta} \in \mathcal{V}_R} z_{\boldsymbol{\beta}} \leq CAR\left(\sqrt{d_{\mathcal{V}} \log d_{\mathcal{V}} d} + \sqrt{\log 2K}\right).$$

$\square$

## C. Proof for Theorem 3.3

**Theorem 3.3.** *The effective dimension of the PI kernel associated with the affine variety* $\mathcal{V}(\mathscr{D}, \mathcal{B}, \mathcal{T}) = \{\boldsymbol{\beta} : \boldsymbol{D}\boldsymbol{\beta} = \boldsymbol{0}\}$ *with dimension* $d_{\mathcal{V}}$ *is upper bounded by*

$$\mathcal{N}(\xi, \nu) \lesssim \sum_{j=1}^{d_{\mathcal{V}}} \frac{1}{1+\xi} + \sum_{j=d_{\mathcal{V}}}^{d} \frac{1}{1+\xi+\nu\lambda_j} \leq \frac{d}{1+\xi}.$$

*where* $\{\lambda_j\}_{j=d_{\mathcal{V}}}^{d}$ *are the eigenvalues of the matrix* $\boldsymbol{D}^{\top}\boldsymbol{G}\boldsymbol{D}$.

*Proof.* From Theorem 4.2 in (Doumèche et al., 2024a) and Equation 15 in (Doumèche et al., 2024b), the effective dimension is bounded as follows:

$$\mathcal{N}(\xi, \nu) \lesssim \sum_{\lambda \in \sigma(\boldsymbol{C}\boldsymbol{M}^{-1}\boldsymbol{C})} \frac{1}{1+\lambda^{-1}} \leq \sum_{\lambda \in \sigma(\boldsymbol{M}^{-1})} \frac{1}{1+\lambda^{-1}}, \tag{20}$$

where $\boldsymbol{M} := \xi\boldsymbol{I} + \nu\boldsymbol{D}^{\top}\boldsymbol{G}\boldsymbol{D} \in \mathbb{R}^{|\mathcal{B}|\times|\mathcal{B}|}$ and $\boldsymbol{C} \in \mathbb{R}^{|\mathcal{B}|\times|\mathcal{B}|}$ is the matrix of the inner products of the basis functions, i.e., $C_{j,j'} = \langle \phi_j, \phi_{j'} \rangle_\mu$ for all $\phi_j, \phi_{j'} \in \mathcal{B}$.

Since the matrix $\boldsymbol{D}^{\top}\boldsymbol{G}\boldsymbol{D}$ is positive semi-definite, the eigenvalues of the matrix $\boldsymbol{M}$ in ascending order $\sigma_j(\cdot)$ are given by

$$\sigma_j(\boldsymbol{M}) = \begin{cases} \xi & (j = 1, \ldots, d_{\mathcal{V}}) \\ \xi + \nu\lambda_j & (d_{\mathcal{V}} < j) \end{cases}.$$

Therefore, the matrix $\boldsymbol{M}$ is positive definite, and the eigenvalues of $\boldsymbol{M}^{-1}$ are $\lambda^{-1}$ for all $\lambda \in \sigma(\boldsymbol{M})$. Combining this with Eq. (20), we obtain the first inequality. The second inequality is obtained when $\nu = 0$. $\square$

## D. Experimental Detail

### D.1. Experiments on Strong Solution

In the experiments in Section 5.1, strong solutions to the equations are obtained analytically. The analytical solution with added Gaussian noise was used as data, the variance of the Gaussian noise was set to 0.01. The hyperparameters $L^2$ regularization weights and differential equation constraint weights $\xi$ and $\nu$ were searched in the range [1e-9, 1e-2] using the Optuna library (Akiba et al., 2019). The configuration with the smallest MSE on the validation data among 100 candidates was selected.

**Harmonic Oscillator:** The initial value problem of a harmonic oscillator $\mathscr{D}[y] = 0$ with spring constant $k$ and mass $m$ on the domain $\Omega = [0, T]$ is given by:

$$\mathscr{D}[y] = \frac{\mathrm{d}^2}{\mathrm{d}t^2}y + \frac{k}{m}y, \ y(0) = y_0, \ \frac{\mathrm{d}}{\mathrm{d}t}y(0) = v_0.$$

We set the parameters $m = k = 1.0$, $T = 2\pi$. The initial position and velocity $[y_0, v_0]^{\top}$ are generated from the normal distribution $\mathcal{N}(\mathbf{1}, I)$, where $\mathbf{1}$ is an all-ones vector and $I$ is the identity matrix. The solution to the initial value problem is analytically given by:

$$y(t) = y_0 \cos(\omega t) + \frac{v_0}{\omega} \sin(\omega t), \ \omega = \sqrt{k/m}.$$

The settings for the basis functions and the test functions with the measure $\phi_j \in \mathcal{B}$, $(\psi_k, \mu_k) \in \mathcal{T}$ are as follows:

$$\phi_1(x) = 1, \ \phi_{2j}(x) = \cos\left(\frac{2\pi j}{T}x\right), \ \phi_{2j+1}(x) = \sin\left(\frac{2\pi j}{T}x\right) \ (j = 1, \ldots, d_t),$$

$$\psi_k(x) = 1, \ \mu_k = \delta_{x_k} \ (k = 1, \ldots, K),$$

where $d_t \in \{2, 4, 8, 16\}$ is the set of the number of basis functions, and $x_k \in \Omega$ is uniformly sampled from data with $K = 100$.

**Diffusion Equation:** The initial value problem for the one-dimensional diffusion equation $\mathscr{D}[u] = 0$ with diffusion coefficient $\alpha$ and periodic boundary conditions is given by:

$$\mathscr{D}[u] = \frac{\partial}{\partial t} u - \alpha \frac{\partial^2}{\partial x^2} u \qquad (x, t) \in [-\Xi, \Xi] \times [0, T]$$
$$u(x, 0) = u_0(x) \qquad\qquad x \in [-\Xi, \Xi]$$
$$u(-\Xi, t) = u(\Xi, t), \quad \frac{\partial u}{\partial x}(-\Xi, t) = \frac{\partial u}{\partial x}(\Xi, t).$$

We set the parameters $\alpha = 1.0$, $\Xi = \pi$, $T = 2\pi$. The initial value $u_0$ is given by:

$$u_0(x) = \sum_{j=0}^{j_{\max}} A_j \cos(\omega_j x) + B_j \sin(\omega_j x), \; \omega_j = \frac{j\pi}{\Xi}, \tag{21}$$

where $[A_j, B_j]^\top$ are generated from the normal distribution $\mathcal{N}(\mathbf{1}, I)$ for all $j = 0, \ldots, j_{\max}$ and $j_{\max}$ is set to 1. The solution to the initial value problem is analytically given by:

$$u(x, t) = \sum_{j=0}^{j_{\max}} [A_j \cos(\omega_j x) + B_j \sin(\omega_j x)] e^{-\alpha \omega_j^2 t}.$$

The settings for the basis functions and the test functions with the measure $\phi_j \in \mathcal{B}$, $(\psi_k, \mu_k) \in \mathcal{T}$ are as follows:

$$\phi_1(x, t) = 1, \; \phi_{2j, j'}(x, t) = \cos(\omega_j x) e^{-\alpha \omega_{j'}^2 t}, \; \phi_{2j+1, j'}(x, t) = \sin(\omega_j x) e^{-\alpha \omega_{j'}^2 t}$$
$$(j = 1, \ldots, d_x, \; j' = 1, \ldots, d_t),$$
$$\psi_k(x, t) = 1, \; \mu_k = \delta_{(x_k, t_k)} \; (k = 1, \ldots, K),$$

where $d_t = 2$, $d_x \in \{10, 15, 20, 25\}$ are the sets of the number of basis functions, and $(x_k, t_k) \in \Omega$ is uniformly sampled from data with $K = 50 \times 500$.

## D.2. Experiments on Numerical Solution

In the experiments in Section 5.2, we numerically simulate the Bernoulli equation using the explicit Euler method and the diffusion equation using the finite difference method (FDM). The data used are the numerical solutions with added Gaussian noise of variance 0.01. The method for hyperparameter search is the same as described in Appendix D.1. For the nonlinear equations, we use the Adam optimizer with a learning rate of $1 \times 10^{-2}$, along with an exponential learning rate scheduler. The training is performed for a maximum of 2000 epochs, utilizing an early stopping technique.

**Discrete Bernoulli Equation:** The discrete Bernoulli equation $\mathscr{D}_h[y] = 0$ with the step size $h$ on the domain $\Omega = [0, T]$ is given by:

$$\mathscr{D}_h[y] = \frac{y_{\tau+1} - y_\tau}{h} + Py_\tau - Qy_\tau^\rho,$$

where $y_\tau = y(t_\tau)$ and $y_{\tau+1} = y(t_\tau + h)$ are evaluations on the grid $\{t_\tau\}_{\tau=1}^{n_t}$ with $n_t = \frac{T}{h}$. We set the constant parameters $(P, Q, \rho)$ to $(1.0, 0.0, 0.0)$ for the linear case and to $(1.0, 0.5, 2.0)$ for the non-linear case. We use varying $n_t \in \{100, 200\}$ with $T = 1.0$ for both cases. The initial state $y_0$ is generated from the standard normal distribution $\mathcal{N}(0, 1)$ for both cases. The ground-truth solution to the initial value problem is numerically solved by the explicit Euler method with step size $h$. The settings for the basis functions and the test functions with measure $\phi_\tau \in \mathcal{B}_h$, $(\psi_\tau, \mu_\tau) \in \mathcal{T}_h$ are as follows:

$$\phi_\tau(t) = \begin{cases} 1 & \text{if } t \in [t_\tau, t_{\tau+1}) \\ 0 & \text{otherwise} \end{cases} \quad (\tau = 1, \ldots, n_t),$$
$$\psi_\tau(t) = \phi_\tau(t), \quad \mu_\tau = \delta_{t_\tau} \quad (\tau = 1, \ldots, n_t),$$

where $n_t = \frac{T}{h}$ is the same as the number of basis and test functions, corresponding to the ground-truth solutions.

**Discrete Diffusion Equation:** The one-dimensional discrete diffusion equation $\mathscr{D}_{\boldsymbol{h}}[u] = 0$ with the step size $\boldsymbol{h} = [h_t, h_x]^\top$ and the diffusion coefficient $\alpha(u)$ on the domain $\Omega = [-\Xi, \Xi] \times [0, T]$ is given by:

$$\mathscr{D}_{\boldsymbol{h}}[u] = \frac{u_j^{\tau+1} - u_j^\tau}{h_t} - \alpha(u_j^\tau) \frac{u_{j+1}^\tau - 2u_j^\tau + u_{j-1}^\tau}{h_x^2},$$

where $u_j^\tau := u(x_j, t_\tau)$, $u_j^{\tau+1} := u(x_j, t_\tau + h_t)$, and $u_{j\pm1}^\tau := u(x_j \pm h_x, t_\tau)$ are evaluations on the $n_x \times n_t$ size grid $\{x_j\}_{j=1}^{n_x} \times \{t_\tau\}_{\tau=1}^{n_t}$, where $n_x := \frac{2\Xi}{h_x}$ and $n_t := \frac{T}{h_t}$. The periodic boundary condition is adopted in the spatial domain, i.e., $u_{n_x+j}^\tau = u_j^\tau$ for any $j \in \mathbb{N}$. The diffusion coefficient $\alpha(u) = 1.0$ is used for the linear case and $\alpha(u) = 0.1/(1+u^2)$ for the nonlinear case. We use varying $(n_t, n_x) \in \{(400, 10), (400, 20), (400, 30)\}$ with $\Xi = 1.0$ and $T = 1.0$ for both cases. The initial value is generated with the same setting as shown in Eq. (21). The ground-truth solution to the initial value problem is numerically solved by the FDM with step sizes $h_t$ for the time domain and $h_x$ for the spatial domain. The settings for the basis functions and the test functions with measure $\phi_{j,\tau} \in \mathcal{B}_{\boldsymbol{h}}$, $(\psi_{j,\tau}, \mu_{j,\tau}) \in \mathcal{T}_{\boldsymbol{h}}$ are as follows:

$$\phi_{j,\tau}(x, t) = \begin{cases} 1 & \text{if } (x, t) \in [x_j, x_{j+1}] \times [t_\tau, t_{\tau+1}] \\ 0 & \text{otherwise} \end{cases} \quad (j = 1, \ldots, n_x, \ \tau = 1, \ldots, n_t),$$

$$\psi_{j,\tau}(x, t) = \phi_{j,\tau}(x, t), \quad \mu_{j,\tau} = \delta_{(x_j, t_\tau)} \quad (j = 1, \ldots, n_x, \ \tau = 1, \ldots, n_t),$$

where $n_x = \frac{2\Xi}{h_x}$ and $n_t = \frac{T}{h_t}$ are the same as the number of basis and test functions, corresponding to the ground-truth solutions.

