# OpenReview forum: "Understanding Generalization in Physics Informed Models through Affine Variety Dimensions"
_ICML.cc/2025/Conference — Submitted to ICML 2025_

### Official Review · Reviewer_r9of · 2025-03-08

**Overall Recommendation:** 1

**Summary:**

First of all, the format of the official review in ICML 2025 is very uncomfortable and would fragment the reviewer's thoughts. My review comments will be summarized in the first block.

This paper theoretically presents the upper bound of the minimax risk of physics-informed linear regressors. The theoretical results provide in-depth discussions on the relation among minimax risk and the intrinsic dimension of the affine variety.

About the novelty and significance. This work focuses on the mechanisms by which and to what extent these physical structures enhance generalization capacity. It is a considerably interesting topic, especially as the hotwave of PINN and the best paper of AAAI 2025 relative to neuro-symbol learning. I have to admit that this focus is also significant, although this paper only takes linear regressors on at most two input dimensions.


About the quality. Unfortunately, I have deep concerns about the quality of this manuscript.


Firstly, this paper is hard to follow. I have to admit that theoretical investigations introduce many uncommon mathematical terminologies and conclusions, inevitably making readers feel bored. However, this manuscript needs to be revised in terms of many points.

On one hand, the introduced terminologies should be tightly tied to machine learning. For example, what does the covering number of balls in hypothesis space mean? Besides, the authors need to be clarified more clearly on what they mean by "physics-informed" in this paper? Does it imply the following assumptions: (1) all basis functions \phi are known, (2) the hypothesis space is separable, i.e., the training error can be reduced to zero, and (3) the hypothesis space can cover the concept or target function, unless I am missing something.

On the other hand, many of the used symbols, especially the subscripts, are unclear. Here, I provide several examples.

1. what are \mu_k, \mathcal{\Tau\}, and \lambda?

2. what are the connection and difference between \phi_j, \boldsymol{phi\}, and \Phi?

3. what is the relation between x and x_i?

4. what is the difference between m and d?
5. what does the "path-connected component" mean?


Secondly, one of the biggest shortcomings of this paper is that it overemphasizes the conclusions without clearly marking the assumptions of this paper. This is actually an academic imprecision. The authors should be asked to provide an obvious form, such as Assumption, to clearly state the assumptions of this paper. Besides, it is better to clarify this sentence: “the generalization capacity is determined by the local size of the hypothesis space induced by the learning algorithm, such as gradient descent.”


Thirdly, in conclusion, the theoretical results of this paper do not bring stronger conclusions or heuristic algorithms, and estimating d_v is not easy.


Fourthly, the design logic of the experiment in this paper is very strange. First, I don't think it is necessary to provide experiments on model performance. Second, for the experiments on generalization theory, the most important thing is to verify the theoretical conclusions of this paper. Unfortunately, this paper always stays in two-dimensional experiments. Finally, if you need to compare model performance, it is unfair to compare PILR with RR. PILR's opponent should be the kernel method.

----
Overall, I admit the interesting topic of this paper; however, the novelty and quality are relatively limited. Therefore, I tend to reject this paper.


## update after rebuttal

The comments are proposed in the reply to the authors' rebuttal or rebuttal comments. Overall, I insist on rejection.

**Claims And Evidence:**

he theoretical results provide in-depth discussions on the relation among minimax risk and the intrinsic dimension of the affine variety. These claims are supported by the informal theorem in Theorem 3.2.

**Essential References Not Discussed:**

na

**Experimental Designs Or Analyses:**

na

**Methods And Evaluation Criteria:**

na

**Other Comments Or Suggestions:**

As summarized in Block 1.

**Other Strengths And Weaknesses:**

As summarized in Block 1.

**Questions For Authors:**

As summarized in Block 1.

**Relation To Broader Scientific Literature:**

na

**Theoretical Claims:**

I did not check the proof in this paper. Limited by the complicated symbols, it would take me twice as long to judge whether the proof in this paper is correct. But based on my experience, the quality of this paper is obviously not worth it. In addition, I can't understand the proof sketch of Theorem 3.2 at all. It seems that the authors is talking to themself.

---

> ### Author Rebuttal · Authors · 2025-03-31
>
> > **Q1.** This paper only takes linear regressors on at most two input dimensions.
>
> **A1.** Our input setting is not limited to two dimensions. As noted in lines 151–168, our analysis handles general $m$-dimensional input, i.e., $x\_i \in \Omega \subset \mathbb{R}^m$.
>
> ---
>
> > **Q2.** What does the covering number of balls in hypothesis space mean?
>
> **A2.** The covering number measures the capacity of a hypothesis space. As this is a standard concept found in basic machine learning textbooks such as [1], we assume readers are familiar with it.
>
> [1] https://cs.nyu.edu/~mohri/mlbook/
>
> ---
>
> > **Q3.** Clarify what "physics-informed" means.
>
> **A3.** As stated on line 22, "physics-informed" refers to models incorporating the structure of differential equations. Specifically, the operator $\mathscr{D}$ is used to constrain the model via $\mathcal{T}$ and $\mathcal{B}$.
>
> ---
>
> > **Q4.** Do the results assume:
> > (1) all basis functions $\phi$ are known,
> > (2) the hypothesis space is separable,
> > (3) it can represent the target function?
>
> **A4.** (1) and (3) are correct, but (2) is not. Due to noise, empirical loss may not reach zero.
> We will revise Sec. 3.2 (around line 149) to state:
> - All basis functions $\phi$ are known.
> - The hypothesis space covers the target function, i.e. $f^* = \beta^* \phi$.
>
> ---
>
> > **Q5.** Many of the used symbols are unclear.
>
> **A5.**  If any parts remain unclear, we would appreciate a more specific explanation of what is confusing.
>
> - $\mu\_k$ is a measure, and $\mathcal{T}$ is a set of test function-measure pairs (lines 119–126).
> - $\phi = (\phi\_1, \dots, \phi\_d): \mathbb{R}^m \rightarrow \mathbb{R}^d$ will be defined explicitly around line 150. $\Phi$, $\phi_j$ are defined in lines 149–150 and 166.
> - $\lambda$ is the eigenvalue of $C M^{-1} C$ (defined in Eq. 10).
> - $x\_i \in \Omega$ denotes a sample, as in lines 154–155. $x$ was a generic element in $\Omega$, but we removed it as unnecessary.
> - $m$ is the input dimension, and $d$ is the number of basis functions/parameters, as evident from the definition of $f^*$, $\mathcal{H}$ and $\beta$.
> - “Path-connected” is a standard concept in math defined as follows.
>   "*$X$ is not path-connected if there exist disjoint open sets $A, B \subset \mathbb{R}^d$ such that $X \subset A \cup B$ and $A \cap X \ne \emptyset$, $B \cap X \ne \emptyset$.*"
>
> ---
>
> > **Q6.** Clarify the sentence: “The generalization capacity is determined by the local size of the hypothesis space ..."
>
> **A6.** Recent work [2][3] shows generalization bounds can depend on data and algorithm choice. These factors effectively restrict the hypothesis space to a smaller, data-dependent subset, enabling tighter bounds. Our sentence highlights this idea and suggests that these aspects offer opportunities for deeper analysis of our bound.
>
> [2] Bartlett, Peter L., et al. "Local Rademacher complexities."
> [3] Steinke, Thomas, et al. "Reasoning about generalization via conditional mutual information."
>
> ---
>
> > **Q7.** Clearly state the assumptions in an explicit form.
>
> **A7.** Due to space limits, Assumptions 1–3 were placed in Appendix B. We will move them to the main text between Lem. 3.1 and Thm. 3.2.
>
> ---
>
> > **Q8.** the theoretical results do not bring stronger conclusions and estimating $d\_V$ is not easy.
>
> **A8.** Our results provide a stronger theoretical contribution by revealing how the structure of general (including **nonlinear**) differential equations affects generalization through affine varieties—an aspect not addressed in prior work. Moreover, the estimation of $d\_V$ is not difficult as described in Sec. 4.2.
>
> ---
>
> > **Q9.** First, it is not necessary to provide experiments on model performance. Second, the most important thing is to verify the theoretical conclusions.
>
> **A9.**  Our experiments are designed to support our primary theoretical claim: a smaller affine variety dimension \( d_V \) leads to better generalization. If there are other experimental setups you consider more appropriate, we would be happy to hear your suggestions.
>
> ---
>
> > **Q10.** Comparing PILR with RR is unfair; kernel methods should be used.
>
> **A10.** We believe the comparison is valid. RR uses feature map $\phi$ and constrains $\beta$ in $\mathbb{B}\_2(R)$, with estimator:
> $$
> \hat{\beta} = (\Phi^{\top}\Phi + n I)^{-1} \Phi y
> $$
> PILR modifies the constraint to $\mathcal{V}\_R$ via differential equations. Comparing RR on $\mathbb{B}\_2(R)$ and PILR on $\mathcal{V}\_R$ is natural. Kernel ridge regression uses RKHS norms, which are not directly comparable.

---

> > ### Comment · Reviewer_r9of · 2025-04-03
> >
> > I have read the rebuttal and other reviewers' comments and would insist on my score, that is, rejection. The reasons are two folds:
> >
> > (1) The theoretical results of this paper do not bring stronger conclusions or heuristic algorithms. I insist on that estimating $d_v$ is not easy in practice since the computation holds on a stronger assumption that physics-informed basis functions are known. Besides, the computational complexity is not discussed.
> >
> >  (2) The presentation is unclear; I believe that it is necessary to provide formal introduction to assumptions, instead of moving to appendix. Emphasizing the conclusion without specifying the conditions or assumptions is like a tree without roots.
> >
> > Overall, I believe that the significance of this work is limited.

---

> > > ### Author Response · Authors · 2025-04-03
> > >
> > > Thank you very much for your feedback. We would like to address the concerns you raised.
> > >
> > > > The theoretical results of this paper do not bring stronger conclusions or heuristic algorithms.
> > >
> > > Could you kindly clarify why our results are considered not stronger, and specify the baseline or prior works you are comparing them to? We believe our theorems provide non-trivial generalization guarantees under a practical hypothesis class and offer meaningful insights.
> > >
> > > ---
> > >
> > > > I insist on that estimating is not easy in practice since the computation holds on a stronger assumption that physics-informed basis functions are known.
> > >
> > > We would like to clarify that the assumption of known physics-informed basis functions and $f^* = \beta^* \phi$ is introduced solely to simplify the presentation and make the theoretical discussion more accessible. Even when this assumption is relaxed, the computation of $d_V$ does not become intractable.
> > >
> > > When $f^* \neq \beta^* \phi$, we can still decompose the excess-risk error as:
> > >
> > > $$
> > > \| f^* - \hat{\beta} \phi \|^2 \leq \| f^* - \beta^* \phi \|^2 + \| \beta^* \phi - \hat{\beta} \phi \|^2,
> > > $$
> > > where $\beta^* := \sum_j \langle f^*, \phi_j \rangle \phi_j$. Our analysis in the main paper can be applied to the second term on the right-hand side in the same way.
> > >
> > > A key point here is that $\beta^* \phi$ must be a weak solution to the differential operator $\mathscr{D}$  (in the sense of Equation (1)). The central question then becomes whether such a basis $\phi$ can be constructed, which we argue is practically feasible.
> > > For example, consider an extreme simple setting: if $\mathcal{T} = \\{ (1, \delta\_0) \\}$, it is sufficient that there exists a $\beta^*$ such that $\mathscr{D}[\beta^*\phi] (0)  = 0$. This condition is mild, and a wide and general class of functions can serve as $\phi$.
> > >
> > > Moreover, even when $\beta^* \phi$ is not an exact weak solution, as long as the violation is small, the increase in the covering number is only linear in the violation. Therefore, the core of our theoretical results remains unchanged.
> > >
> > > ---
> > >
> > > > Besides, the computational complexity is not discussed.
> > >
> > > For the linear case, computing $d_V$ reduces to calculating the rank of a matrix of size $|\mathcal{T}| \times d$. Using standard algorithms, this requires:
> > >
> > > $$
> > > O(\min(|\mathcal{T}|, d) \cdot |\mathcal{T}| \cdot d)
> > > $$
> > >
> > > In the nonlinear case, we perform this computation over $N$ samples, which leads to a total computational cost of:
> > >
> > > $$
> > > O(N \cdot \min(|\mathcal{T}|, d) \cdot |\mathcal{T}| \cdot d)
> > > $$
> > >
> > > The computational complexity is practical and feasible in most scenarios considered in our setting.
> > >
> > > ---
> > >
> > > > The presentation is unclear; I believe that it is necessary to provide formal introduction to assumptions, instead of moving to appendix. Emphasizing the conclusion without specifying the conditions or assumptions is like a tree without roots.
> > >
> > > We would like to respectfully clarify that our paper does not omit the assumptions entirely from the main text, nor do we attempt to obscure them in any way. In fact, the assumptions are clearly stated in the main body of the paper in the following form:
> > >
> > > * *"We only concern the estimation error by assuming $f^* = \beta^* \phi$." (line 193-195)
> > > * *“Suppose that the basis function is bounded by a constant, the minimum eigenvalue of the design matrix is restricted, and the stability condition for the estimator holds.”* (line 184-187)
> > >
> > > These are the all mathematical assumptions underlying our theoretical results. The precise formal statements, including the full notation and constants, were moved to the appendix solely due to space constraints. Therefore, the analogy that our conclusions are “like a tree without roots” does not accurately reflect the structure of our presentation.
> > >
> > > **We fully agree with the importance of presenting assumptions clearly, and we move the formal version of these assumptions to the main text in the final version of the paper. Since the final version allows for one additional page, we will ensure these details are included in a clear and visible manner.**
> > >
> > > We hope this addresses the concern and assures the reviewer that our presentation remains mathematically faithful and transparent.

---

### Official Review · Reviewer_5pFq · 2025-03-12

**Overall Recommendation:** 3

**Summary:**

This paper focuses on analyzing the generalization ability of physics-informed machine learning models. It shows that for linear regressors with differential equation structures, the generalization performance is determined by the dimension of the associated affine variety instead of the number of parameters. The authors conduct a minimax risk analysis, introduce a method to approximate the dimension of the affine variety, and provide experimental evidence. The results demonstrate that physical structures can reduce the intrinsic dimension of the hypothesis space and prevent overfitting.

=============

I read the response and other reviews, and keep my score unchanged.

**Claims And Evidence:**

Yes, they are all supported by solid evidence.

**Essential References Not Discussed:**

NA

**Experimental Designs Or Analyses:**

There’s no significant problem in the experimental design or analysis of this paper.

**Methods And Evaluation Criteria:**

Correct as the analyzing methods used in this work are, they seem to be not technically novel. Those bounds to the generation loss derived by the methods are also trivial.

**Other Comments Or Suggestions:**

NA

**Other Strengths And Weaknesses:**

Strengths:
1. Theoretical Innovation: The paper proposes a novel approach to analyze the generalization ability of physics - informed models by using the dimension of affine varieties. This new perspective provides a unified theoretical framework for understanding these models, which is a significant contribution to the field.
2. Strong Experimental Validation: The authors conduct a series of experiments, comparing ridge regression and physics - informed linear regression for various equations. These experiments, under different data sizes and numbers of parameters, effectively verify the theoretical analysis.

Weaknesses:
1. Limited Model Types: The analysis in this paper is confined to linear models. It does not extend to more complex deep learning models like Physics-Informed Neural Networks (PINNs). This limitation restricts the universality of the research findings.

**Questions For Authors:**

1. The bound given by theorem 3.2 is trivial, and it can be derived without the analyzing tools used, which means it isn’t of too much theoretical value. Could you give a more tight bound on the minimax risk?
2. Could you take a step further to identify which property PINN possesses enables good generalizing capacity?

**Relation To Broader Scientific Literature:**

The paper aligns with the growing body of work in PIML(Physical Informed Machine Learning). However, this work fails to give explicit insight about what a good PINN architecture should be like, or to identify the inherent properties of PINN which enable it to generalize better theoretically. Instead, the paper just give a few general bounds, which are even trivial. Therefore, the contributions of the paper to the PIML or PINN fields are not that significant.

**Theoretical Claims:**

The theoretical claims in this work have all been strictly proved.

---

> ### Author Rebuttal · Authors · 2025-03-30
>
> > Q1. The bound given by Theorem 3.2 is trivial and can be derived without the analyzing tools used. Could you give a tighter bound on the minimax risk?
>
> Our generalization bound cannot be derived without the analyzing tools we introduced, particularly the covering number of the affine variety. Our bound is sufficiently tight to explain the performance gap between the physics-informed and non-physics-informed linear regressors.
>
> While it is true that tighter bounds may be obtained in future work, e.g., by considering algorithm-dependent generalization bounds [1], such bounds mix the effects of the hypothesis space and the learning algorithm. This makes it difficult to isolate and analyze the contribution of the physical structure. Therefore, while tightness is desirable, it does not necessarily lead to better understanding of the underlying mechanisms.
>
> [1] Steinke, Thomas, and Lydia Zakynthinou. "Reasoning about generalization via conditional mutual information." Conference on Learning Theory. PMLR, 2020. (https://proceedings.mlr.press/v125/steinke20a.html)
>
> > Q2. Could you take a step further to identify which property PINN possesses enables good generalizing capacity?
>
> As stated in the conclusion, our analysis is limited to linear models, and extending it to neural networks is left as future work. On the practical side, our contribution lies in determining the appropriate size of $|\mathcal{T}|$ (the number of PDE evaluation points in the PINN setting) via $d_V$ for each equations, which is a nontrivial insight. Moreover, reliability guarantees in numerical analysis are valuable in engineering applications in their own right. These contributions are expected to carry over when extended to more general neural networks.

---

### Official Review · Reviewer_CVSb · 2025-03-14

**Overall Recommendation:** 2

**Summary:**

This paper provides an analytical framework for the generalization of linear regressors that incorporate differential equation structures. The authors demonstrate that the generalization bound depends on the dimension of the associated affine variety rather than the number of parameters. Additionally, they show that their theory aligns with existing work on physics-informed (PI) kernels when the operator is linear.

**Claims And Evidence:**

The theoretical claims are proven and supported by experimental results.

**Essential References Not Discussed:**

I cannot confidently identify essential references that are missing from the paper's discussion.

**Experimental Designs Or Analyses:**

The experimental designs appear sound and provide appropriate validation of the theoretical results.

**Methods And Evaluation Criteria:**

The methods and evaluation criteria are appropriate for addressing the research questions posed in the paper.

**Other Comments Or Suggestions:**

No additional comments.

**Other Strengths And Weaknesses:**

Strengths:

1. The paper is well-written with clear logical flow, making complex theoretical concepts accessible.
2. The paper deals with an important problem in the field of theory of physics-informed modeling.

Weaknesses:

1. In the proof of Theorem 3.2, Assumption 2 requires the minimum singular value of $\Phi$ to be positive, which typically holds only when $n = d$. This raises questions about how to interpret the number of parameters in the more general case.

2. The novelty appears somewhat limited. The key lemma was previously developed by Zhang & Kileel (2023), and Theorem 3.2 seems to be largely an application of this result. Please correct me if I have missed any important technical contributions.

3. The concept that the dimension of the affine variety for physics-informed models affects generalization performance is conceptually similar to how the rank of the sample matrix affects generalization in standard linear models. Therefore, the results developed in this paper in some sense are expected for physics-informed models.

**Questions For Authors:**

1. Does the generalization bound depend on the number of test functions, i.e., $|\mathcal{T}|$? If so, how does this relationship manifest?
2. Regarding Theorem 3.3, could you clarify how the upper bound of the effective dimension of the PI kernel becomes smaller when the dimension of the affine variety decreases?
3. In Figures 2(a) and 2(b), the test loss appears to increase with the number of parameters for some models. Does this observation contradict your theory that generalization depends on the dimension of the affine variety rather than the number of parameters?
4. In the experiments presented in Tables 1 and 2, is it possible to design experiments that fix $d$ while varying $d_v$ to directly demonstrate how test loss changes with different values of the affine variety dimension?

**Relation To Broader Scientific Literature:**

This paper extends generalization bounds to nonlinear differential equations, an area not previously well-addressed in the literature. The contribution has potential foundational value for understanding the generalization performance of physics-informed models more broadly.

**Theoretical Claims:**

I reviewed the proofs and found them to be logically sound, though I did not verify every mathematical step in extensive detail.

---

> ### Author Rebuttal · Authors · 2025-03-30
>
> > Q1. Assumption 2 requires positive min. singular value of X, which typically holds only if n ≥ d...
>
> A1. Since $\hat{\beta} - \beta^* \in \mathbb{B}_2(2R)$, Eq. (12) still holds under a weaker condition. For $\kappa > 0$,
> $$
> \frac{1}{\sqrt{n}}\\|\Phi\beta\\| \geq \sqrt{\kappa} \\|\beta\\|_2,\quad \forall \beta \in \mathbb{B}_2(2R).
> $$
>
> If $R$ is sufficiently small, it is reasonably expected that $\beta$ will not lie in the eigenspace corresponding to the eigenvalue 0 of $\Phi$. We will reflect this relaxation in the manuscript.
>
> ---
>
> > Q2. The key lemma is from Zhang & Kileel (2023); Thm 3.2 seems like a direct application...
>
> A2. While we rely on the covering number bound from Zhang & Kileel (2023), applying this result to the analysis of physics-informed models is itself a nontrivial contribution. Our contribution lies in formulating the generalization problem from the viewpoint of affine varieties—a perspective not previously explored in the ML or PIML literature.
>
> ---
>
> > Q3. Affine variety dim. vs. generalization resembles rank-based arguments...
>
> A3. This is an insightful point. However, the design matrix $\Phi$ in our setting is not inherently low-rank. The low-rank property *emerges* by imposing the structure of the differential equations. One of the main contributions can be viewed as reducing the behavior of physics-informed models to rank-based arguments—a nontrivial step that helps uncover their underlying structure.
>
> In addition, rank-based analysis is limited to linear differential equations. Our use of the affine variety dimension generalizes this to nonlinear cases, offering a more broadly applicable perspective in ML and physics-informed contexts.
>
> ---
>
> > Q4. Does the gen. bound depend on $|\mathcal{T}|$? If so, how...
>
> A4. Yes. $|\mathcal{T}|$ determines the number of constraints. As $|\mathcal{T}|$ increases, the variety $\mathcal{V}$ becomes smaller:
>
> $$
> \mathcal{T}\_1 \subset \mathcal{T}\_2 \Rightarrow \mathcal{V}(\mathscr{D},\mathcal{B},\mathcal{T}\_2) \subset \mathcal{V}(\mathscr{D},\mathcal{B},\mathcal{T}\_1) \Rightarrow d\_{\mathcal{V}(\mathscr{D},\mathcal{B},\mathcal{T}\_2)} \leq d\_{\mathcal{V}(\mathscr{D},\mathcal{B},\mathcal{T}\_1)}.
> $$
> We will add this clarification around lines 168–178.
>
> ---
>
> > Q5. Clarify why Thm 3.3 upper bound shrinks as affine dim. decreases...
>
> A5. Since $D^\top G D$ is positive semi-definite ($\lambda_j \geq 0$), we have $1/(1+\xi) > 1/(1+\xi\lambda_j)$. As $d_V$ decreases, more terms in the bound use $1/(1+\xi\lambda_j)$ instead of $1/(1+\xi)$, leading to a smaller total. We will clarify this and emphasize the semi-definite property of $G$ in Definition 3.1 and Theorem 3.3.
>
> ---
>
> > Q6. In Fig. 2, test loss increases with parameters in some cases... Contradiction?
>
> A6.  There is no contradiction. Our theory provides an upper bound, and fluctuations in performance within that bound are expected. Additionally, the test loss may be influenced by the choice of the hyperparameter $\lambda$. Furthermore, our bound includes a $\log d$ term in the first component, so it is not entirely independent of $d$.
>
> ---
>
> > Q7. Can you design experiments varying $d_V$ while fixing $d$?
>
> A7. Yes. By varying $|\mathcal{T}|$, we control $d_V$ independently of $d$, and we provide results based on this setting.
>
> **Linear Bernoulli**
>
> | $d$  | $d\_{\mathcal{V}}$ | Test MSE |
> |------|-------------------|----------|
> | 101  | 10                | $0.012 \pm 0.0023$ |
> | 101  | 20                | $0.125 \pm 0.0817$ |
> | 101  | 40                | $0.329 \pm 0.2160$ |
>
> **Nonlinear Bernoulli**
>
> | $d$  | $d\_{\mathcal{V}}$ | Test MSE |
> |------|-------------------|----------|
> | 100  | 10                | $0.170 \pm 0.1094$ |
> | 100  | 20                | $0.206 \pm 0.1409$ |
> | 100  | 40                | $0.329 \pm 0.2259$ |
>
> **Linear Heat**
>
> | $d$   | $d\_{\mathcal{V}}$ | Test MSE |
> |-------|-------------------|----------|
> | 4010  | 110               | $1.64 \pm 0.35$ |
> | 4010  | 210               | $1.88 \pm 0.44$ |
> | 4010  | 410               | $2.03 \pm 0.49$ |
>
> **Nonlinear Heat**
>
> | $d$   | $d\_{\mathcal{V}}$ | Test MSE |
> |-------|-------------------|----------|
> | 2010  | 110               | $0.366 \pm 0.113$ |
> | 2010  | 210               | $0.430 \pm 0.141$ |
> | 2010  | 410               | $0.557 \pm 0.188$ |
>
> **Observation:**
> Across all settings, test MSE increases as the dimension $d_{\mathcal{V}}$ grows.

---

### Decision · Program_Chairs · 2025-05-01

**Decision:**

Reject

**Comment:**

This paper examines the generalization performance of physics-informed models, deriving a generalization bound which is shown to depend on the dimension of an associated affine variety rather than the number of parameters. Empirical analysis corroborates the theoretical results in a few settings.

The reviewers agree that the problem is salient and that the experimental results convincingly verify the theoretical analysis. On the other hand, they also note that the paper is hard to follow, the novelty may be somewhat limited, and that the theory does not necessarily provide strong conclusions or paths towards new algorithms. Having said that, I'd acknowledge that developing a generalization bound for nonlinear differential equations does offer the potential for new directions in statistical learning theory. Still, the reviewer concerns suggest that the current version of the draft is not appropriate for publication at ICML. I'd encourage the authors to address any reviewer comments and pursue an alternate venue.